# QUANTUM SPEEDUPS IN LINEAR PROGRAMMING VIA SUBLINEAR MULTI-GIBBS SAMPLING

## ABSTRACT

As a basic optimization technique, linear programming has found wide applications in many areas. In this paper, we propose an improved quantum algorithm for solving a linear programming problem with $m$ constraints and $n$ variables in time $\widetilde{O}(\sqrt{m+n}\gamma^{2.25})$, where $\gamma = Rr/\varepsilon$ is the additive error $\varepsilon$ scaled down with bounds $R$ and $r$ on the size of the primal and dual optimal solutions, improving the prior best $\widetilde{O}(\sqrt{m+n}\gamma^{2.5})$ by Bouland, Getachew, Jin, Sidford, and Tian (2023) and Gao, Ji, Li, and Wang (2023). Our algorithm solves linear programming via a zero-sum game formulation, under the framework of the sample-based optimistic multiplicative weight update. At the heart of our construction, is an improved quantum multi-Gibbs sampler for diagonal Hamiltonians with time complexity *sublinear* in inverse temperature $\beta$, breaking the general $O(\beta)$-barrier.

## 1    INTRODUCTION

Linear programming is a powerful mathematical optimization technique used to achieve the best outcomes given a set of requirements represented as linear relationships. It involves maximizing a linear objective function subject to linear equality or inequality constraints, where the variables can take real values. It enables optimal and efficient decision-making in many complex real-world problems, including resource allocation, portfolio optimization, and network flows. It is also fundamental to combinatorial optimization.

To be more concrete, a linear programming (LP) problem can be formulated in the following standard form: consider a matrix $\mathbf{A} \in \mathbb{R}^{m \times n}$; the question is to decide the maximum value of $c^{\mathsf{T}}x$ where $c$ is a given vector in $\mathbb{R}^n$ and $x$ is the variable vector under the constraints $Ax = b$ for some $b \in \mathbb{R}^m$. In the study of general LP algorithms, people usually assume $m = \Theta(n)$ for simplicity. Also, there are cases where the algorithm can only output an approximate optimal value with $\varepsilon$ additive error.

In 1947, the simplex method was proposed by George Dantzig to solve the linear programming problem. The simplex algorithm is very efficient in practice. However, in 1972, it was shown that the worst time complexity for the simplex algorithm is exponential (with respect to $n$) (Klee & Minty, 1972). The linear programming problem was proven to be solvable in polynomial time by Leonid Khachiyan in 1979 (Khachiyan, 1980).

In recent years, there have been several works (Cohen et al., 2021; Jiang et al., 2021) which aim to give faster algorithms for the linear programming problem. In Cohen et al. (2021), the authors gave an algorithm for LP in $O^*(n^\omega)$ time[1] for current value $\omega \approx 2.37$ and $O^*(n^{2+1/6})$ if $\omega = 2$. In Cohen et al. (2021), the authors improve the latter complexity to $O^*(n^{2+1/18})$ if $\omega \approx 2$.

Linear programming has deep connections to matrix games in game theory. A matrix game involves two players who each have to choose between a finite set of pure strategies. The payoffs for each player are given in a payoff matrix based on the pure strategies chosen. Matrix games can be expressed as linear programs, with the payoff matrix providing the coefficients for the objective function and constraints. The linear programming formulation allows for identifying the optimal mixed strategies that maximize the expected payoff for each player. Solving the corresponding dual linear program yields the value of the game. Therefore, techniques developed for solving linear programs can also be applied to finding optimal solutions for matrix games. Conversely, finding optimal solutions for matrix games can also be transformed into algorithms for linear programming (Vaserstein, 2006).

In Grigoriadis & Khachiyan (1995), the authors gave an $\widetilde{O}((n+m)/\varepsilon^2)$ time randomized algorithm for matrix games. This is done by updating a single entry of the strategy vectors each time. In Carmon et al. (2019), using the variance reduction technique, the authors gave a $\widetilde{O}(mn + \sqrt{mn(m+n)}/\varepsilon)$ time algorithm for matrix games.

---

[1]Throughout this paper, $\omega \approx 2.37$ denotes the matrix multiplication exponent.

Quantum computation utilizes the principle of quantum mechanics to perform computation in a different way from classical computation. Rather than using classical binary bits (which only have two states 0 and 1), quantum computers use quantum bits (qubits) to store data and perform computation. The state of qubits can be in a superposition of 0 and 1 states. Harnessing this superposition feature, quantum algorithms are able to achieve speedups than their classical counterparts. For instance, Shor's algorithm for integer factorization and Grover's algorithm for database search provide provable speedups over the best-known classical algorithms.

Over the years, many quantum algorithms have been developed with quantum speedups over their classical counterparts. In the area of optimization, quantum algorithms for semi-definite programming are one of the most important research areas in quantum algorithms (Brandão & Svore, 2017; van Apeldoorn et al., 2017; Brandão et al., 2019; van Apeldoorn & Gilyén, 2019a). Other examples of quantum algorithms include quantum recommendation systems (Kerenidis & Prakash, 2017) and quantum algorithms for training linear and kernel-based classifiers (Li et al., 2019).

In van Apeldoorn et al. (2017), the authors pointed out that their algorithm for semi-definite programming can also be applied to solve linear programming problems in time $\widetilde{O}(\sqrt{mn}(Rr/\varepsilon)^5)$ where $R$ and $r$ are parameters related to the numerical scale of the problem. Then, in van Apeldoorn & Gilyén (2019b), the authors gave a quantum algorithm specifically for matrix zero-sum games and linear programming with running time $\widetilde{O}(\sqrt{m+n}(Rr/\varepsilon)^3)$, which is by designing a quantum Gibbs sampler and use the framework proposed in Grigoriadis & Khachiyan (1995). Following this work, in Bouland et al. (2023), the authors proposed an improved dynamic Gibbs sampler, which results in an $\widetilde{O}(\sqrt{m+n}/\varepsilon^{2.5})$-time solver for matrix zero-sum games.

This work presents improved quantum algorithms for matrix zero-sum games and linear programming by extending the framework of sample-based optimistic multiplicative weight update first proposed by Gao et al. (2023). The framework requires a specific task called multi-Gibbs sampling, which requires the quantum subroutine to collect multiple samples from the Gibbs distribution in a single update iteration. In their work, they used the "preparing many copies of a quantum state" technique of Hamoudi (2022) and the quantum singular value transformation (Gilyén et al., 2019) to give an efficient quantum multi-Gibbs sampler. All the previous quantum Gibbs samplers used in van Apeldoorn & Gilyén (2019b); Bouland et al. (2023); Gao et al. (2023) have a linear dependence on the $\ell_1$-norm $\beta$ of the vector $u$. The parameter $\beta$ plays the role of inverse temperature as it scales the diagonal Hamiltonian $H = \mathrm{diag}(u)/\beta$ of trace norm 1. This $\beta$-dependence is known as the $\Omega(\beta)$ barrier and is proved for general quantum Gibbs sampling (Gilyén et al., 2019; Wang & Zhang, 2023). Surprisingly, our improved multi-Gibbs sampler breaks this $\Omega(\beta)$ bound in the sense of amortized complexity per sample under certain conditions. This improvement is the key to our further speedup compared with the previous approach in Gao et al. (2023). By combining this new multi-Gibbs sampler with the sample-based optimistic weight update framework, we present an $\widetilde{O}(\sqrt{m+n}/\varepsilon^{2.25})$-time quantum solver for matrix zero-sum games and $\widetilde{O}(\sqrt{m+n}(Rr/\varepsilon)^{2.25})$-time quantum linear programming solver.

## 1.1 OUR RESULT

We propose quantum algorithms for solving matrix zero-sum games and linear programming problems, which improves on the aspect of the runtime of the prior state-of-the-art quantum algorithms (Bouland et al., 2023; Gao et al., 2023).

**Theorem 1.1** (Informal version of Corollary 4.2). *There exists a quantum algorithm that, for $\varepsilon \in (0, 1/2)$ satisfying $1/\varepsilon = \widetilde{O}((m+n)^2)$, returns an $\varepsilon$-approximate Nash equilibrium for the zero-sum game $\mathbf{A} \in \mathbb{R}^{m \times n}$ with probability at least $2/3$ in $\widetilde{O}(\sqrt{m+n}/\varepsilon^{2.25})$ time.*

Notice that our theorem requires $1/\varepsilon = \widetilde{O}((m+n)^2)$. If this does not hold, i.e., $1/\varepsilon = \widetilde{\Omega}((m+n)^2)$, we can directly uses the algorithm in Grigoriadis & Khachiyan (1995) with better time complexity.

In comparison, the algorithms in Bouland et al. (2023); Gao et al. (2023) with time complexity $\widetilde{O}(\sqrt{m+n}/\varepsilon^{2.5})$ requires $1/\varepsilon = O((m+n)^{-1})$. Our algorithm allows a wider range of parameter choices and achieves further quantum speedups on this problem.

For the linear programming solver, we have:

**Theorem 1.2** (Informal version of Corollary 4.3). *There exists a quantum algorithm that, for $\varepsilon \in (0, 1/2)$, returns an $\varepsilon$-feasible and $\varepsilon$-optimal solution for linear programming problems of $n$ variables and $m$ constraints with probability at least $2/3$ in $\widetilde{O}(\sqrt{m+n}\gamma^{2.25})$ time, provided that $R$, $r$ are the bounds on the $\ell_1$ norm of the primal and dual optimal solutions and $\gamma = Rr/\varepsilon = \widetilde{O}((m+n)^2)$.*

In Table 1, we compare our algorithm with previous classical and quantum algorithms for linear programming.

The essential part in proving our theorem is an improved quantum multi-Gibbs sampler for diagonal Hamiltonians which we state below.

**Theorem 1.3** (Informal version of Theorem 3.2). *There exists a quantum algorithm such that, for every $\varepsilon \in (0, 1/2)$ and a $(\beta, \lceil \log_2(n) \rceil, O(1), O(1))$-amplitude-encoding $V$ of a vector $u \in \mathbb{R}^n_{\geq 0}$ with $\beta \geq 1$, if the number of copies $k$ satisfies $k = \widetilde{\Omega}(\sqrt{\beta})$ and $k = \widetilde{O}(n\sqrt{\beta})$, then with probability at least $1 - \varepsilon$ the algorithm will return $k$ samples from a distribution that is $\varepsilon$-close to the Gibbs distribution of $u$ in the total variation distance in $\widetilde{O}(\beta^{3/4}\sqrt{nk})$ time.*

Table 1: Classical and quantum linear programming solvers

| Method | Approach | Type | Classical/Quantum Time Complexity |
|---|---|---|---|
| Multiplicative Weight | Grigoriadis & Khachiyan (1995) | Classical | $\widetilde{O}((m + n)\gamma^2)$ |
| Multiplicative Weight | Syrgkanis et al. (2015) | Classical | $\widetilde{O}(mn\gamma)$ |
| Multiplicative Weight | Li et al. (2019) | Quantum | $\widetilde{O}(\sqrt{m + n}\gamma^4)$ |
| Multiplicative Weight | van Apeldoorn & Gilyén (2019b) | Quantum | $\widetilde{O}(\sqrt{m + n}\gamma^3)$ |
| Multiplicative Weight | Bouland et al. (2023) | Quantum | $\widetilde{O}(\sqrt{m + n}\gamma^{2.5})$ |
| Multiplicative Weight | Gao et al. (2023) | Quantum | $\widetilde{O}(\sqrt{m + n}\gamma^{2.5})$ |
| Multiplicative Weight | Our result | Quantum | $\widetilde{O}(\sqrt{m + n}\gamma^{2.25})$ |
| Interior Point | Jiang et al. (2021) | Classical | $O^*((m + n)^\omega)$ |
| Interior Point | Casares & Martin-Delgado (2020) | Quantum | $\widetilde{O}(\sqrt{n}(m + n)M\kappa/\varepsilon^2)$ [†] |

[†] $M$ and $\kappa$ are the Frobenius norm and condition number of the systems of linear equations in the algorithm.

## 1.2 MAIN TECHNIQUES

Quantum Gibbs sampling was used in solving SDP and LP problems (Brandão & Svore, 2017; van Apeldoorn et al., 2017; Brandão et al., 2019; van Apeldoorn & Gilyén, 2019a;b; Bouland et al., 2023). Recently, a specifically designed Gibbs sampling for diagonal Hamiltonians was proposed in van Apeldoorn & Gilyén (2019b) for solving zero-sum games and LPs. In van Apeldoorn & Gilyén (2019b), their quantum Gibbs sampler adopts the idea of quantum rejection sampling (Grover, 2000; Ozols et al., 2013). The sampler first prepares a uniform superposition state $|\Psi\rangle$, and then applies a unitary block-encoding of $\exp(\beta H)$ on the state, where $H$ is the diagonal Hamiltonian $\mathrm{diag}(u - u_{\max})/\beta$ where $u = \mathbf{A}x$ with $\beta \geq \|u\|_1$, resulting in a state

$$|\psi\rangle \approx \frac{1}{\sqrt{n}}|0\rangle|u_{\mathrm{Gibbs}}\rangle + |1\rangle|\mathrm{garbage}\rangle,$$

where measuring $|u_{\mathrm{Gibbs}}\rangle$ in the computational basis will return a classical sample from the desired Gibbs distribution. The unitary block-encoding is constructed by quantum singular value transformation (QSVT) with a polynomial approximating $\exp(\beta x)$ of degree $\widetilde{O}(\beta)$.

In Gao et al. (2023), they improved the procedure of van Apeldoorn & Gilyén (2019b) by (i) preparing a non-uniform initial state $|\Psi'\rangle$ after a $\widetilde{O}(\beta\sqrt{nk})$-time pre-processing procedure, adapting from the "preparing many copies of a quantum state" techniques in Hamoudi (2022), (ii) applying a unitary block-encoding of $\exp(\beta H')$ where $H'$ is determined by the pre-processing procedure, which results in a state

$$|\psi'\rangle \approx \sqrt{\frac{k}{n}}|0\rangle|u_{\mathrm{Gibbs}}\rangle + |1\rangle|\mathrm{garbage}\rangle.$$

Then, they can obtain a copy of $|u_{\mathrm{Gibbs}}\rangle$ in $\widetilde{O}(\beta\sqrt{n/k})$ time, thereby $k$ copies in $\widetilde{O}(\beta\sqrt{nk})$.

In this paper, we further improved the multi-sampling of Gao et al. (2023) with two novel observations.

**Better polynomial approximation.** In the previous works (van Apeldoorn & Gilyén, 2019b; Bouland et al., 2023; Gao et al., 2023), they used a polynomial of degree $\widetilde{O}(\beta)$ to approximate the function $\exp(\beta x)$. We observe that the polynomial is only required to be well-behaved on the interval $[-1, 0]$. Thus, the polynomial approximation for $\exp(-\beta - \beta x)$ suffices, with a polynomial of degree $\widetilde{O}(\sqrt{\beta})$ known in Sachdeva & Vishnoi (2014). Using this polynomial, we can reduce the time complexity of step (ii) of the algorithm of Gao et al. (2023) to $\widetilde{O}(\sqrt{\beta nk})$ for preparing $k$ copies of $|u_{\mathrm{Gibbs}}\rangle$.

**Tradeoff between pre-processing and multi-sampling.** Even though we reduce the time complexity of step (ii) of the algorithm of Gao et al. (2023), the overall time complexity still remains unchanged due to the dominating time complexity of the pre-processing procedure. We thus parameterize both the pre-processing and multi-sampling procedures with time complexity $\widetilde{O}(\beta\sqrt{n\xi})$ and $\widetilde{O}(k\sqrt{\beta n/\xi})$, respectively. As a result, the time complexity is reduced successfully as

$$\widetilde{O}(\beta\sqrt{n\xi}) + \widetilde{O}(k\sqrt{\beta n/\xi}) = \widetilde{O}(\beta^{3/4}\sqrt{nk})$$

by setting $\xi = \widetilde{\Theta}(k/\sqrt{\beta})$.

## 2 PRELIMINARIES

### 2.1 NOTATIONS

Through this paper, we will fix these notations: $[n]$ stands for the set $\{1,\ldots,n\}$. The symbol $\mathbb{R}^n_{\geq 0}$ stands for the set of $n$-dimensional vectors with non-negative entries. The symbol $\Delta_n$ stands for the probability simplex $\left\{x \in \mathbb{R}^n_{\geq 0} : \sum_{i=1}^n x_i = 1\right\}$. For a vector $u \in \mathbb{R}^n$, the notation $\text{diag}(u)$ stands for the diagonal matrix in $\mathbb{R}^{n\times n}$ with diagonal entries being the entries of $u$.

### 2.2 QUANTUM COMPUTATION

**Input Model.** For matrix zero-sum games and linear programming, the input will be a matrix $\mathbf{A} = (A_{i,j})_{i,j=1}^n \in \mathbb{R}^{m\times n}$. However, the time will be $\Omega(mn)$ if all the entries of $\mathbf{A}$ are read classically. Thus, in classical literature Grigoriadis & Khachiyan (1995), they assume an oracle access $f_\mathbf{A}(\cdot,\cdot)$ to $\mathbf{A}$. The oracle function $f_\mathbf{A}$ acts as follows: $f_\mathbf{A}(i,j) = A_{i,j}$. Following this idea, previous quantum algorithms for zero-sum games (van Apeldoorn & Gilyén, 2019b; Bouland et al., 2023; Gao et al., 2023) used a quantum analog of this oracle, namely a unitary $\mathcal{O}_\mathbf{A}$ which acts as follows:

$$\mathcal{O}_\mathbf{A}|i\rangle|j\rangle|k\rangle = |i\rangle|j\rangle|k \oplus A_{i,j}\rangle.$$

Here, we assume that $A_{i,j}$ has a finite floating number precision. We also assume that we have an oracle access to the unitary $\mathcal{O}'_\mathbf{A}$ that satisfies:

$$\mathcal{O}'_\mathbf{A}|i\rangle|j\rangle|0\rangle = |i\rangle|j\rangle \otimes \left(\sqrt{A_{i,j}}|0\rangle + \sqrt{1 - A_{i,j}}|1\rangle\right).$$

**QRAM.** Quantum-read classical-write random access memory (QRAM) is a common assumption in many quantum algorithms. The memory can store classical data, and it allows superposition query access. In previous quantum algorithms for linear programming problems (van Apeldoorn & Gilyén, 2019b; Bouland et al., 2023; Gao et al., 2023), they all utilize the QRAM to achieve quantum read-access for efficiently constructing unitaries.

**Complexity Measure.** For the query complexity of quantum algorithms, when we claim we use queries to $U$, we mean that we use queries to $U$, controlled-$U$, and their inverses. For the time complexity, following Apers & de Wolf (2022), we say a quantum algorithm has time complexity $T$, if it uses at most $T$ one- and two-qubit gates, quantum queries to the input, and QRAM operations.

### 2.3 BASIC QUANTUM ALGORITHMS

**Quantum Minima Finding.** Finding the minimal $k$ elements in a database with $n$ entries is a common task. It is known in Dürr et al. (2006) that quantum algorithms have quadratic speedups not only over $n$ but also over $k$. Here, we state a modified version of their theorem for our use, which aims to find maximal elements instead of minimal ones.

**Theorem 2.1** (Quantum maxima finding, Adapted from (Dürr et al., 2006, Theorem 3.4)). *Given $k \in [n]$, and quantum oracle $\mathcal{O}_a$ for an array $a_1, a_2, \ldots, a_n$, i.e., $\mathcal{O}_a: |i\rangle|0\rangle \mapsto |i\rangle|a_i\rangle$ for all $i \in [n]$, there is a quantum algorithm $\mathsf{FindMax}(\mathcal{O}_a, n, k, \varepsilon)$ that, with probability at least $1 - \varepsilon$, finds a set $S \subseteq [n]$ of cardinality $|S| = k$ such that $a_i \geq a_j$ for all $i \in S$ and $j \notin S$, using $O(\sqrt{nk}\log(1/\varepsilon))$ queries to $\mathcal{O}_a$ and in $O(\sqrt{nk}\log(n)\log(1/\varepsilon))$ time.*

**Quantum Amplitude Amplification.** The procedure of quantum amplitude amplification is a generalization of the Grover search, and it is commonly used in the context of quantum singular value related algorithms for amplifying a desired state. Here, we state the theorem for our later use.

**Theorem 2.2** (Adapted from (Brassard et al., 2002, Theorem 3)). *Let $U$ be an $n \times n$ unitary matrix. Suppose that $U|0\rangle|0\rangle = \sqrt{p}|0\rangle|\phi_0\rangle + \sqrt{1-p}|1\rangle|\phi_1\rangle$, where $p \in (0,1)$, $|\phi_0\rangle$ and $|\phi_1\rangle$ are normalized pure quantum state. There exists a quantum algorithm $\mathsf{Amp}(U, \varepsilon)$, such that, with probability at least $1 - \varepsilon$, output the state $|\phi_0\rangle$, using $O(\log(1/\varepsilon)/\sqrt{p})$ queries to $U$ and in $O(\log(n)\log(1/\varepsilon)/\sqrt{p})$ time.*

**Consistent Quantum Amplitude Estimation.** Besides quantum amplitude amplification, amplitude estimation is also a very useful quantum procedure. However, the result of the quantum amplitude estimation usually depends on the measurement. In our algorithm, we need a consistent version of amplitude estimation, which is stated as follows:

**Theorem 2.3** (Adapted from Gao et al. (2023, Theorem C.3)). *Let $U$ be an $n \times n$ unitary matrix. Suppose that: $U|0\rangle|0\rangle = \sqrt{p}|0\rangle|\phi_0\rangle + \sqrt{1-p}|1\rangle|\phi_1\rangle$, where $p \in (0,1)$, $|\phi_0\rangle$ and $|\phi_1\rangle$ are normalized pure quantum states. Then there exists a quantum algorithm $\mathsf{AmpEst}(U, s, \delta)$ such that, on input $\delta > 0$ and an $O(r)$-bit random string $s$, the algorithm outputs $f(s, p)$ with probability at least $1 - \exp(-\Omega(r))$ such that $|f(s, p) - p| \leq \delta$, using $O(r/\delta)$ queries to $U$ and in $O(r \log(n)/\delta)$ time.*

## 2.4 QUANTUM SINGULAR VALUE TRANSFORMATION

Quantum singular value transformation is a powerful quantum algorithm design framework proposed in Gao et al. (2023). Here we review some key concepts and theorems which will be used later for our algorithm design.

**Block-Encoding.** The concept of block-encoding is fundamental to the quantum singular value transformation framework. The definition of block-encoding is as follows:

**Definition 2.1.** *Suppose $A$ is a linear operator on a Hilbert space of $s$ qubits. For an $(s + a)$-qubit unitary operator $U$, we call it an $(\alpha, a, \varepsilon)$-block-encoding of $A$, if $U$ satisfies $\|A - \alpha\langle 0|^{\otimes a} U |0\rangle^{\otimes a}\| \leq \varepsilon$.*

**Scaling Technique for Block-Encoding Operators.** Sometimes the coefficient $\alpha$ in the block-encoding is a barrier for later constructions of the entire algorithm. Thus we need the following lemma for adjusting the coefficients of block-encoding operators.

**Lemma 2.4** (Up-scaling of block-encoded operators, (Wang & Zhang, 2023, Corollary 2.8)). *Suppose that unitary operator $U$ is a $(1, a, \varepsilon)$-block-encoding of $A/\alpha$ with $\|A\| \leq 1$. Then, there is a quantum circuit $\mathsf{BlockAmp}(U, \alpha)$ that is a $(1, a + 2, 8\alpha\varepsilon)$-block-encoding of $A$, using $O(\alpha)$ queries to $U$ and in $O((a + 1)\alpha)$ time.*

**Linear Combination of Unitaries.** Linear combination of unitaries (LCU) is a powerful technique to use existing block-encodings of some linear operators to obtain block-encodings for the linear combination of these operators. To state the lemma clearly, we first need the definition of the state preparation pair.

**Definition 2.2** (State preparation pair, (Gilyén et al., 2019, Definition 28)). *Let $y \in \mathbb{R}^n$ be an $m$-dimensional vector; in this context, we require the coordinate index to start at $0$, and $\|y\|_1 \leq \beta$ for some $\beta > 0$. The unitary pair $(P_L, P_R)$, both acting on $b$ qubits, is called a $(\beta, b, \varepsilon)$-state-preparation-pair for $y$, if*

$$P_L|0\rangle^{\otimes b} = \sum_{i=0}^{2^b-1} c_j|j\rangle, \quad P_R|0\rangle^{\otimes b} = \sum_{i=0}^{2^b-1} d_j|j\rangle,$$

*such that $\sum_{j=0}^{m-1} |y_j - \beta c_j^* d_j| \leq \varepsilon$, and for $j = m, \ldots, 2^b - 1$, $c_j^* d_j = 0$.*

Now we can state the LCU lemma as follows:

**Lemma 2.5** (Linear combination of block-encoded matrices, (Gilyén et al., 2019, Lemma 29)). *Let $\{A_j\}_{j=0}^{m-1}$ be a set of linear operators of the same dimension. For all $j \in \{0, 1, \ldots, m-1\}$, suppose we have $U_j$ which is a $(\alpha, a, \varepsilon_1)$-block-encoding of $A_j$. For an $m$-dimensional vector $y \in \mathbb{R}^m$, suppose $\beta \geq \|y\|_1$ and $(P_L, P_R)$ is a $(\beta, b, \varepsilon_2)$-state preparation pair for $y$. Define $A = \sum_{j=0}^{m-1} y_j A_j$ and*

$$W = \sum_{j=0}^{m-1} |j\rangle\langle j| \otimes U_j + \left( I - \sum_{j=0}^{m-1} |j\rangle\langle j| \right) \otimes I_a \otimes I_s.$$

*Then, we can implement a unitary $\mathsf{LCU}((U_j)_{j=0}^{m-1}, P_L, P_R)$ which is an $(\alpha\beta, a + b, \alpha\beta\varepsilon_1 + \alpha\varepsilon_2)$-block-encoding of $A$, using $O(1)$ queries to $P_L, P_R$ and $W$.*

**Polynomial Eigenvalue Transformation.** We are now ready to state polynomial eigenvalue transformation, which is a special case of quantum singular value transformation when we have a block-encoding of a Hermitian matrix. The result of the polynomial eigenvalue transformation is obtained by combining this special case of the general QSVT theorem and the LCU lemma. The theorem can be stated as follows:

**Theorem 2.6** (Gilyén et al. (2019, Theorem 31)). *Suppose a unitary operator $U$ is an $(\alpha, a, \varepsilon)$-block-encoding of a Hermitian matrix $A$. For every $\delta > 0$ and real polynomial $P(x) \in \mathbb{R}[x]$ of degree $d$, satisfying $\sup_{x \in [-1,1]} |P(x)| \leq \frac{1}{2}$, there is a quantum circuit $\mathsf{EigenTrans}(U, P, \delta)$ which is a $(1, a + 2, 4d\sqrt{\varepsilon/\alpha} + \delta)$-block-encoding of $P(A/\alpha)$. The circuit consists of $O(d)$ queries to $U$, and $O((a + 1)d)$ other one- and two-qubit gates. Moreover, the description of the quantum circuit can be computed in $O(\mathrm{poly}(d, \log(1/\delta)))$ time on a classical computer.*

### 2.5 POLYNOMIAL APPROXIMATION RESULTS FOR QSVT

**Chebyshev Polynomial.** We define Chebyshev polynomial $T_d(x)$ of degree $|d|$ for every integer $d$ by $T_d(x) = 2xT_{d-1}(x) - T_{d-2}(x)$ for $d \geq 2$ with $T_0(x) = 1$ and $T_1(x) = x$. We also denote $T_d(x) = T_{|d|}(x)$ for $d < 0$.

**Polynomial Approximation of Monomials.** In Sachdeva & Vishnoi (2014), they show that a degree $d$ monomial with coefficient 1 can be approximated by a polynomial of degree $\sqrt{d}$. The exact statement is as follows:

**Theorem 2.7** (Sachdeva & Vishnoi (2014, Theorem 3.3)). *For positive integers $s$ and $d$, let $p_{s,d}(x) = \sum_{i=-d}^{d} \frac{1}{2^s} \binom{s}{\frac{s+i}{2}} T_i(x)$ be a polynomial of degree $d$, where $\binom{n}{m} = 0$ if $m$ is not an integer between $0$ and $n$. Then, $\sup_{x \in [-1,1]} |p_{s,d}(x) - x^s| \leq 2 \exp(-d^2/2s)$.*

**Polynomial Approximation of Exponential Functions.** Using the above result and the Taylor expansion of exponential functions, we have the following theorem for the approximation of exponential functions.

**Theorem 2.8** (Sachdeva & Vishnoi (2014, Lemma 4.2)). *For every $\lambda > 0$, $\delta \in (0, 1/2]$, we choose $t = O(\lambda + \log(\delta^{-1}))$ and $d = O(\sqrt{t \log(\delta^{-1})})$ and define polynomial $q_{\lambda,t,d}(x) = \exp(-\lambda) \sum_{i=0}^{t} \frac{(-\lambda)^i}{i!} p_{i,d}(x)$ of degree $d$. Then, $\sup_{x \in [-1,1]} |q_{\lambda,t,d}(x) - \exp(-\lambda - \lambda x)| \leq \delta$.*

### 2.6 SAMPLERTREE

**SamplerTree.** The SamplerTree is a quantum data structure that combines binary tree and QRAM characteristics to efficiently construct unitaries. See Kerenidis & Prakash (2017); Gilyén et al. (2019) for more discussions. We have the following lemma for describing the functionality of the SamplerTree data structure.

**Lemma 2.9** (Adapted from Kerenidis & Prakash (2017, Theorem 5.1) and Gilyén et al. (2019, Lemma 48 in the full version)). *Let $u \in \mathbb{R}^n_{\geq 0}$ be a vector. There is a data structure $\mathsf{SamplerTree}$, of which an instance $\mathcal{T}$ can maintain the vector $u$ and support the following operations:*

- *Initialization: $\mathsf{SamplerTree.Initialize}(n, c)$: return an instance of the $\mathsf{SamplerTree}$, and set $u_i \leftarrow c$ for all $i \in [n]$ in this instance, where $c \geq 0$, in $O(1)$ time.*

- *Assignment $\mathcal{T}.\mathsf{Assign}(i, c)$: set $u_i \leftarrow c$ for some index $i$, where $c \geq 0$, in $O(\log(n))$ time.*

- *State Preparation: output a unitary $\mathcal{T}.\mathsf{Prepare}(\varepsilon)$ which satisfies:*

$$\left\| \mathcal{T}.\mathsf{Prepare}(\varepsilon)|0\rangle - \sum_{i=1}^{n} \sqrt{\frac{u_i}{\|u\|_1}} |i\rangle \right\| \leq \varepsilon,$$

  *in $O(\log^2(n) \log^{5/2}(n\|u\|_1/\varepsilon))$ time.*

- *Query Access: output a unitary $\mathcal{T}.\mathsf{BlockEnc}(\varepsilon)$ where $\beta \geq \max_i |u_i|$ which is a $(1, O(1), \varepsilon)$-block-encoding of $\mathrm{diag}(u/\beta)$ in $O(\log(n) + \log^{5/2}(\beta/\varepsilon))$ time.*

### 2.7 QUANTUM ACCESS TO CLASSICAL DATA

**Amplitude-Encoding.** The concept of amplitude-encoding is proposed in Gao et al. (2023). This concept is a way to specify how classical data is stored and accessed in quantum computation.

**Definition 2.3.** *Let $V$ be an $(a + b + c)$-qubit unitary operator acting on subsystems $A, B, C$ with $a, b, c$ qubits, respectively. $V$ is said to be a $(\beta, a, b, c)$-amplitude-encoding of a vector $u \in \mathbb{R}_{\geq 0}^n$ with $a \geq \log_2(n)$, if for all $i \in [n]$, the following holds:*

$$\langle 0|_C V|0\rangle_C |i\rangle_A |0\rangle_B = \sqrt{\frac{u_i}{\beta}} |i\rangle_A |\psi_i\rangle_B,$$

*where $|\psi_i\rangle$ is a normalized pure state. When $a, b, c$ are not important or explicit in the context, we simply call $V$ a $\beta$-amplitude-encoding of $u$.*

The following lemma shows that we can transform the amplitude-encoding to block-encoding.

**Lemma 2.10** (Adapted from Gao et al. (2023, Proposition D.11))**.** *Let $V$ be a $(\beta, a, b, c)$-amplitude-encoding of a vector $u \in \mathbb{R}_{\geq 0}^n$. Then there is an algorithm $\mathsf{AmpToBlock}(V)$ which returns (the classical description of) a $(\beta, b + 2c, 0)$-block-encoding of $\mathrm{diag}(u)$, using $O(1)$ queries to $V$.*

## 3 MULTI-GIBBS SAMPLING ALGORITHM

In this part, we propose an improved algorithm for the multi-Gibbs sampling task. For readability, we first introduce a pre-processing procedure in the first subsection and then introduce the main algorithm which uses the pre-processing algorithm as a subroutine.

### 3.1 PRE-PROCESSING

The pre-processing algorithm is shown in Algorithm 1, and its correctness and complexity are analyzed in Theorem 3.1. The idea of this algorithm is to use consistent amplitude estimation to access the classical data in the amplitude-encoding, then use the quantum maximum finding algorithm to find the largest $\ell$ elements for the later state preparation procedure. It should be noted that amplitude estimation could only return an estimate rather than the exact value. Thus, we can only guarantee that our maximum finding can only find the largest $\ell$ elements of the estimate rather than the true value.

---

**Algorithm 1** $\mathsf{GibbsPre}(V, \ell, \varepsilon)$: pre-processing of the multi-Gibbs sampling

**Input:** Failure probability parameter $\varepsilon$, a $(\beta, \lceil \log(n) \rceil, O(1), O(1))$-amplitude-encoding $V$ of a vector $u \in \mathbb{R}_{\geq 0}^n$, and $\ell \in [n]$.
**Output:** A set $S \subseteq [n]$, and $\widetilde{u}_i$'s for all $i \in S$.
  1: Generate a $\Theta(\log(n\ell/\varepsilon))$-bit random string $s$.
  2: $S \leftarrow \mathsf{FindMax}(\mathsf{AmpEst}(V', s, 1/2\beta), n, \ell, \varepsilon/2)$, where

$$V' = (\mathsf{XOR}_{D,C})^\dagger (V \otimes I_D)(\mathsf{XOR}_{D,C}).$$

  3: **for** all $i \in S$ **do**
  4:    Prepare the state $\mathsf{AmpEst}(V', s, 1/2\beta)|i\rangle|0\rangle$ and measure the last register.
  5:    Store the measurement result classically as $\widetilde{u}_i$.
  6: **end for**
  7: Output the set $S$ and $\widetilde{u}_i$'s for $i \in S$.

---

**Theorem 3.1.** *For every $\ell \in [n]$, $\varepsilon \in (0, 1/2)$, and $(\beta, \lceil \log_2(n) \rceil, O(1), O(1))$-amplitude-encoding $V$ of a vector $u \in \mathbb{R}_{\geq 0}^n$, algorithm $\mathsf{GibbsPre}(V, \ell, \varepsilon)$ (Algorithm 1) outputs the following with probability at least $1 - \varepsilon$*

  - *a set $S$ such that there exist $\widetilde{u}_i$'s satisfying $u_i \leq \widetilde{u}_i \leq u_i + 1$ for all $i \in [n]$, and $S$ contains the indices of the largest $\ell$ elements of $\widetilde{u}_i$.*

  - *a list of non-negative real numbers $\widetilde{u}_i$'s for all $i \in S$,*

*using $O(\beta \sqrt{n\ell} \log(n\ell/\varepsilon) \log(1/\varepsilon))$ queries to $V$, in time $O(\beta \sqrt{n\ell} \log(n\ell/\varepsilon) \log(1/\varepsilon) \log(n))$.*

### 3.2 SAMPLING

In the following, we will present the main multi-Gibbs sampling algorithm. In the algorithm, we will need two fixed unitary matrices $P_L, P_R$ as a state-preparation-pair for the linear combination of unitaries. These matrices should be

chosen to satisfy the following requirements:

$$P_L|0\rangle = \frac{1}{\sqrt{6}}|0\rangle - \frac{1}{\sqrt{6}}|1\rangle - \frac{\sqrt{2}}{\sqrt{3}}|2\rangle, \quad P_R|0\rangle = \frac{1}{2}|0\rangle + \frac{1}{2}|1\rangle + \frac{1}{2}|2\rangle + \frac{1}{2}|3\rangle. \tag{1}$$

---

**Algorithm 2** $\mathsf{Gibbs}(V, k, \varepsilon)$: Gibbs sampling

---

**Input:** A $(\beta, \lceil \log(n) \rceil, O(1), O(1))$-amplitude-encoding $V$ of the vector $u \in \mathbb{R}^n_{\geq 0}$, sample count $k$, and $\varepsilon \in (0, 1/2)$.
**Output:** Samples $i_1, i_2, \ldots, i_k$.
1: Compute $\ell = \left\lfloor \frac{k \log(k/\varepsilon)}{\beta^{1/2} \log^{1/2}(n/\varepsilon) \log(1/\varepsilon)} \right\rfloor$;
2: Compute $\varepsilon_q = \Theta(\ell \varepsilon^2 / n)$, and $\varepsilon_2 = \Theta(\varepsilon_q^2 / d^2)$;
3: Compute $t = \Theta(\beta + \log(\varepsilon_q^{-1}))$, and $d = \Theta(\sqrt{t \log(\varepsilon_q^{-1})})$.
4: $(S, (\widetilde{u}_i)_{i \in S}) \leftarrow \mathsf{GibbsPre}(V, \ell, \varepsilon/2)$;
5: Compute $\widetilde{u}_{\min} = \min_{i \in S} \widetilde{u}_i$;
6: Compute $W = (n - \ell) \exp(\widetilde{u}_{\min}) + \sum_{i \in S} \exp(\widetilde{u}_i)$;
7: $\mathcal{T} \leftarrow \mathsf{SamplerTree.Initialize}(n, \widetilde{u}_{\min})$;
8: $\mathcal{T}_{\exp} \leftarrow \mathsf{SamplerTree.Initialize}(n, \exp(\widetilde{u}_{\min})/W)$;
9: **for** all $i \in S$ **do**
10: $\quad \mathcal{T}.\mathsf{Assign}(i, \widetilde{u}_i)$;
11: $\quad \mathcal{T}_{\exp}.\mathsf{Assign}(i, \exp(\widetilde{u}_i)/W)$;
12: **end for**
13: Compute the classical description of $U_{\mathsf{LCU}} = \mathsf{LCU}((\mathcal{T}.\mathsf{BlockEnc}(\varepsilon_2), \mathsf{AmpToBlock}(V), I), P_L, P_R)$, where $P_L$ and $P_R$ are defined in Equation (1);
14: Compute the classical description of $U_{\mathsf{ET}} = \mathsf{EigenTrans}(\mathsf{BlockAmp}(U_{\mathsf{LCU}}, \sqrt{6}), q_{\beta, t, d}, \varepsilon_q/2)$;
15: **for** $l = 1, \ldots, k$ **do**
16: $\quad$ Prepare the state $|\psi_l\rangle = \mathsf{Amp}(U_{\mathsf{ET}} \cdot (I \otimes \mathcal{T}_{\exp}.\mathsf{Prepare}(\varepsilon_q)), \varepsilon/2k)$;
17: $\quad$ Measure $|\psi_l\rangle$ in the computational basis, and store the outcome as $i_l$;
18: **end for**
19: Output $i_1, \ldots, i_k$.

---

We have the following theorem for the algorithm:

**Theorem 3.2.** *For every $\varepsilon \in (0, 1/2)$, integer $k > 0$, and a $(\beta, \lceil \log_2(n) \rceil, O(1), O(1))$-amplitude-encoding $V$ of a vector $u \in \mathbb{R}^n_{\geq 0}$ with $\beta \geq 1$, if*

$$1 \leq \frac{k \log(k/\varepsilon)}{\beta^{1/2} \log^{1/2}(n/\varepsilon) \log(1/\varepsilon)} \leq n,$$

*then with probability at least $1 - \varepsilon$, Algorithm 2 will return $k$ samples from a distribution that is $\varepsilon$-close to the Gibbs distribution of $u$, using*

$$Q_{\mathrm{Gibbs}}(n, k, \beta, \varepsilon) = O\left( \sqrt{nk} \left( \beta^{3/4} + \beta^{1/4} \log^{1/2}(n/\varepsilon) \right) \log^{1/2}(1/\varepsilon) \log^{3/4}(n/\varepsilon) \log^{1/2}(k/\varepsilon) \right)$$

*queries to $V$, and in*

$$T_{\mathrm{Gibbs}}(n, k, \beta, \varepsilon) = O\left( \frac{k \log(k/\varepsilon) \log(n)}{\beta^{1/2} \log^{1/2}(n/\varepsilon) \log(1/\varepsilon)} + Q_{\mathrm{Gibbs}}(n, k, \beta, \varepsilon) \log^2(n) \log^{2.5}(n\beta/\varepsilon) \right)$$

*time.*

# 4 COMPUTING THE NASH EQUILIBRIUM OF ZERO-SUM GAMES

In this section, we discuss applying our multi-Gibbs sampling procedure to computing the $\varepsilon$-approximate Nash equilibrium of two-person zero-sum games.

## 4.1 THE SETUP

The problem setting is as follows: suppose we are given a matrix $\mathbf{A} \in \mathbb{R}^{m \times n}$ with entries $a_{i,j} \in [0, 1]$. The goal of our algorithm is to find the approximate optimal strategies $x \in \Delta_m$, $y \in \Delta_n$, such that $\max_{y' \in \Delta_n} x^\mathsf{T} \mathbf{A} y' - \min_{x' \in \Delta_m} x'^\mathsf{T} \mathbf{A} y \leq \varepsilon$.

## 4.2 QUANTUM OPTIMISTIC MULTIPLICATIVE WEIGHT UPDATE

---

**Algorithm 3** Quantum Optimistic Multiplicative Weight Update

---

**Input:** Quantum oracle to the element $a_{i,j}$ of the matrix $\mathbf{A} \in \mathbb{R}^{m \times n}$, step size $\lambda$, additive approximation error $\varepsilon$, total round $T$.

**Output:** The $\varepsilon$-approximate Nash equilibrium strategy pair $(u, v)$.

1: Set $u \leftarrow \mathbf{0}_m$, $v \leftarrow \mathbf{0}_n$, $\zeta^{(0)} \leftarrow \mathbf{0}_m$, $\eta^{(0)} \leftarrow \mathbf{0}_n$, $x^{(0)} \leftarrow \mathbf{0}_m$, and $y^{(0)} \leftarrow \mathbf{0}_n$.
2: **for** $t = 1, \ldots, T$ **do**
3:     $(i_1^{(t)}, i_2^{(t)}, \ldots, i_T^{(t)}) \leftarrow \mathsf{Gibbs}(-\mathbf{A}y^{(t-1)}, T, \varepsilon_G)$.
4:     $(j_1^{(t)}, j_2^{(t)}, \ldots, j^{(t)}) \leftarrow \mathsf{Gibbs}(\mathbf{A}^\mathsf{T}x^{(t-1)}, T, \varepsilon_G)$.
5:     $\zeta^{(t)} \leftarrow \sum_{l=1}^{T} e(i_l^{(t)})/T$.
6:     $\eta^{(t)} \leftarrow \sum_{l=1}^{T} e(j_l^{(t)})/T$.
7:     $x^{(t)} \leftarrow x^{(t-1)} + 2\zeta^{(t)} - \zeta^{(t-1)}$.
8:     $y^{(t)} \leftarrow y^{(t-1)} + 2\eta^{(t)} - \eta^{(t-1)}$.
9:     $u \leftarrow u + \zeta^{(t)}$.
10:    $v \leftarrow v + \eta^{(t)}$.
11: **end for**
12: **return** the pair $(u/T, v/T)$.

---

In Gao et al. (2023), the authors proved the following theorem for Algorithm 3.

**Theorem 4.1** (Gao et al. (2023, Theorem 3.2))**.** *Suppose $T = \Theta(\log(mn)/\varepsilon)$, $\varepsilon_G = O(\varepsilon/\log(mn))$, and $\lambda \in (0, \sqrt{3}/6)$ be a constant. Then with probability at least $2/3$, Algorithm 3 will return an $\varepsilon$-approximate Nash equilibrium for the zero-sum game $\mathbf{A}$.*

Using our Theorem 3.2, we have the following corollary:

**Corollary 4.2.** *There exists a quantum algorithm that, for $\varepsilon \in (0, 1/2)$, with probability at least $2/3$, returns an $\varepsilon$-approximate Nash equilibrium for the zero-sum game $\mathbf{A} \in \mathbb{R}^{m \times n}$, using*

$$O(\mathrm{Q}_{\mathrm{Gibbs}}(m + n, \Theta(\log(mn)/\varepsilon), \Theta(\log(mn)/\varepsilon), \varepsilon^3)) = \widetilde{O}(\sqrt{m + n}/\varepsilon^{9/4})$$

*queries to A, and in*

$$O(\mathrm{T}_{\mathrm{Gibbs}}(m + n, \Theta(\log(mn)/\varepsilon), \Theta(\log(mn)/\varepsilon), \varepsilon^3)) = \widetilde{O}(\sqrt{m + n}/\varepsilon^{9/4})$$

*time, provided that $1/\varepsilon = \widetilde{O}((m + n)^2)$.*

## 4.3 APPLICATION: LINEAR PROGRAM SOLVER

As is discussed in van Apeldoorn & Gilyén (2019b); Gao et al. (2023), solving linear programs can be reduced to finding an $\varepsilon$-approximate Nash equilibrium of a related zero-sum game. See Appendix D for a more detailed discussion of the reduction. Thus, we have the following corollary:

**Corollary 4.3.** *There exists a quantum algorithm that, for $\varepsilon \in (0, 1/2)$, with probability at least $2/3$, returns an $\varepsilon$-feasible and $\varepsilon$-optimal solution for the linear programming problem:*

$$
\begin{aligned}
\underset{x \in \mathbb{R}^n}{\text{minimize}} \quad & c^\mathsf{T}x \\
\text{subject to} \quad & \mathbf{A}x \leq b, \\
& x \geq 0
\end{aligned}
\tag{2}
$$

*which uses*

$$O(\mathrm{Q}_{\mathrm{Gibbs}}(m + n, \Theta(\log(mn)Rr/\varepsilon), \Theta(\log(mn)Rr/\varepsilon), (\varepsilon/Rr)^3)) = \widetilde{O}(\sqrt{m + n}(Rr/\varepsilon)^{9/4})$$

*queries to $\mathbf{A}$, $b$, and $c$, and runs in*

$$O(\mathrm{T}_{\mathrm{Gibbs}}(m + n, \Theta(\log(mn)Rr/\varepsilon), \Theta(\log(mn)Rr/\varepsilon), (\varepsilon/Rr)^3)) = \widetilde{O}(\sqrt{m + n}(Rr/\varepsilon)^{9/4})$$

*time, provided that $Rr/\varepsilon = \widetilde{O}((m + n)^2)$.*

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
