# A  CONSISTENT AMPLITUDE ESTIMATION

In this subsection, we review the consistent phase estimation and amplitude estimation procedure and give a proof for Theorem 2.3. We first recall the phase estimation procedure.

**Theorem A.1.** *There exists a quantum algorithm* $\mathsf{PhaseEst}(U, \varepsilon, \delta)$ *such that, for* $\delta \in (0, 1)$ *and* $\varepsilon \in (0, 1)$, *and an* $n \times n$ *unitary matrix* $U$ *such that*

$$U = \sum_{j \in [n]} \exp(2\pi \mathrm{i} \lambda_j) |v_j\rangle \langle v_j|,$$

*where* $\lambda_j \in [0, 1)$, *it holds that:*

- *On input any state* $\sum_{j \in [n]} a_j |v_j\rangle |0\rangle$, *the algorithm outputs a state* $\sum_{j \in [n]} a_j |v_j\rangle |\tilde{\varphi}_j\rangle$.

- *If measuring* $|\tilde{\varphi}_j\rangle$ *on the computational basis, then with probability at least* $1 - \varepsilon$ *we will get the approximation* $\tilde{\varphi}_j$ *that satisfies* $|\tilde{\varphi}_j - \lambda_j| \leq \delta$.

*Moreover, the algorithm uses* $O(\varepsilon^{-1} \delta^{-1})$ *queries to* $U$ *and in* $O(\varepsilon^{-1} \delta^{-1})$ *time. gates*

To describe and use the quantum phase estimation procedure more precisely, following Ta-Shma (2013), we introduce the following notations:

Denote $t$ to be the number of ancilla qubits and $T = 2^t$. Then define

$$\mathrm{Far}_{\delta, T}(\lambda) := \left\{ j \in \mathbb{Z} \cap [0, T-1] : \left| \left( \frac{j}{T} - \lambda \right) \bmod 1 \right| \geq \delta \right\}$$

Using the above notation, we can write the quantum phase estimation algorithm (without measurement) to be the following unitary:

$$\mathsf{PhaseEst}(U, \varepsilon, \delta) \colon \sum_{j \in [n]} a_j |v_j\rangle |0\rangle \mapsto \sum_{j \in [n]} a_j |v_j\rangle |\tilde{\varphi}_j\rangle,$$

where

$$|\tilde{\varphi}_j\rangle = \sum_{k=0}^{T-1} \beta_{j,k} |k\rangle,$$

and

$$\sum_{k \in \mathrm{Far}_{\delta, T}(\lambda_j)} |\beta_{j,k}|^2 \leq \varepsilon.$$

In the consistent phase estimation algorithm, we need to divide the interval $[s\delta' - \delta', 1 + s\delta' + \delta')$ into consecutive sections of length $\delta$. The following function indicates which intersection the estimation result falls in:

$$\mathrm{Sec}(x) = \left\lfloor \frac{x - s\delta' + \delta'}{\delta} \right\rfloor.$$

Let $\mathsf{Sec}$ be the associated unitary of this function that works as follows:

$$\mathsf{Sec} = I \otimes \sum_{k} \left( |k\rangle \langle k| \otimes \sum_{l} |l + \mathrm{Sec}(k/T)\rangle \langle l| \right).$$

We are now ready to state the consistent phase estimation algorithm with a theorem for its correctness.

**Theorem A.2** (Adapted from Ta-Shma (2013)). *Let* $U$ *be an* $n \times n$ *unitary matrix such that*

$$U = \sum_{j \in [n]} \exp(2\pi \mathrm{i} \lambda_j) |v_j\rangle \langle v_j|, \tag{3}$$

*where* $\lambda_j \in [0, 1)$. *Let* $V \subseteq [n]$, *and* $S = \{\lambda_j : j \in V\}$ *with cardinality* $|S|$. *There exists a quantum algorithm, namely Algorithm 4, such that for* $\delta \in (0, 1)$ *and* $\varepsilon \in (0, 1)$, *on input* $O(\log(|S|\varepsilon^{-1}))$-*bit random string* $s$, *it holds with probability at least* $1 - \varepsilon$ *that:*

---

**Algorithm 4** Consistent Phase Estimation

---

**Input:** A unitary $U$ of the form Equation (3), $\delta > 0$, and $\varepsilon > 0$. State $|\psi\rangle = \sum_{j \in V} a_j |v_j\rangle$ where $V \subseteq [n]$. A random string $s$ with $\lceil \log(\lceil 2|S|/\varepsilon \rceil) \rceil$ bits, where $|S|$ is the cardinality of the set $S = \{\lambda_j : j \in V\}$.

**Output:** A state that is $O(\sqrt{\varepsilon})$-close to the state $\sum_{j \in V} a_j |v_j\rangle |f(s, \lambda_j)\rangle$ in trace distance, where $f(s, \lambda)$ is the estimation result.

1: Set $\zeta = \varepsilon/|S|$, $\delta' = \zeta\delta/2$, $L = \lceil \delta/\delta' \rceil$.
2: Treat $s$ as a random number in $\{0, \ldots, L-1\}$. Modify the unitary $U$ as $U' = \exp(2\pi i s \delta') U$.
3: Divide the interval $[s\delta' - \delta', 1 + s\delta' + \delta']$ into consecutive sections of length $\delta$.
4: Prepare $(\mathsf{PhaseEst}(U', \varepsilon, \delta'))^\dagger \cdot \mathsf{Sec} \cdot \mathsf{PhaseEst}(U', \varepsilon, \delta')|\psi\rangle|0\rangle|0\rangle$.
5: Discard the second to last register and output the state.

---

- *On input any state $|\psi\rangle = \sum_{j \in V} a_j |v_j\rangle$, the algorithm outputs a state that is $O(\sqrt{\varepsilon})$-close to the state $\sum_{j \in V} a_j |v_j\rangle |f(s, \lambda_j)\rangle$ in trace distance,*

*where $f(s, \lambda)$ is a function only of $s$ and $\lambda$ such that $|f(s, \lambda) - \lambda| \leq \delta$. Moreover, the algorithm uses $O(|S|\varepsilon^{-2}\delta^{-1})$ queries to $U$ and in $O(|S|\varepsilon^{-2}\delta^{-1})$ time.*

*Proof.* We know for

$$|\psi\rangle = \sum_{j \in V} a_j |v_j\rangle,$$

we have:

$$\mathsf{Sec} \cdot \mathsf{PhaseEst}(U', \varepsilon, \delta')|\psi\rangle|0\rangle|0\rangle = \mathsf{Sec} \cdot \mathsf{PhaseEst}(U', \varepsilon, \delta') \left( \sum_{j \in V} a_j |v_j\rangle |0\rangle |0\rangle \right)$$

$$= \mathsf{Sec} \left( \sum_{j \in V} \sum_{k=0}^{T-1} a_j \beta_{j,k} |v_j\rangle |k\rangle |0\rangle \right)$$

$$= \sum_{j \in V} \sum_{k=0}^{T-1} a_j \beta_{j,k} |v_j\rangle |k\rangle |\mathrm{Sec}(k/T)\rangle.$$

Let $\lambda$ be an eigenvalue of $U$. After shifting, the eigenvalue becomes $\lambda + s$. Consider the interval division in Algorithm 4; suppose a typical section is of the form $[c, d)$ where $d = c + \delta$. If $\lambda + s$ falls in some section $[c, d)$ with the additional property that $c + \delta' \leq \lambda + s < d - \delta'$, then we say $s$ is a good shift for $\lambda$. Equivalently, this means $c \leq \lambda + s - \delta'$ and $\lambda + s + \delta' < d$. Thus, we find that for all $j \notin \mathrm{Far}_{\delta', T}(\lambda)$, $j/T$ will fall into the same section $[c, d)$. Denote the result as $f(s, \lambda)$.

So, if $s$ is a good shift for all eigenvalues in $S$, we have:

$$\sum_{j \in V} \sum_{k=0}^{T-1} a_j \beta_{j,k} |v_j\rangle |k\rangle |\mathrm{Sec}(k/T)\rangle$$

$$= \sum_{j \in V} a_j |v_j\rangle \left( \sum_{j \notin \mathrm{Far}_{\delta', T}(\lambda)} \beta_{j,k} |k\rangle |f(s, \lambda_j)\rangle + \sum_{j \in \mathrm{Far}_{\delta', T}(\lambda)} \beta_{j,k} |k\rangle |\mathrm{Sec}(k/T)\rangle \right).$$

Thus, the state has fidelity at least $1 - \varepsilon$ with the following state:

$$\sum_{j \in V} a_j |v_j\rangle \otimes \left( \sum_{k=0}^{T-1} \beta_{j,k} |k\rangle \right) \otimes |f(s, \lambda_j)\rangle = \mathsf{PhaseEst}(U', \varepsilon, \delta') \left( \sum_{j \in V} a_j |v_j\rangle \otimes |0\rangle \otimes |f(s, \lambda_j)\rangle \right).$$

Then after reversing the phase estimation procedure, i.e., applying $(\mathsf{PhaseEst}(U', \varepsilon, \delta'))^\dagger$, the fidelity between two states will be still at least $1 - \varepsilon$. Thus, we get a state that is $O(\sqrt{\varepsilon})$-close to the state $\sum_{j \in V} a_j |v_j\rangle |f(s, \lambda_j)\rangle$ in trace distance.

Now, we compute the probability that the particular shift $s$ we choose is a good shift. Notice that $L\delta' < \delta$, we know that for every eigenvalue $\lambda \in S$, there exists at most 2 shifts that are not good. Thus the probability that $s$ is not good for an eigenvalue in $S$ is no more than $2/L \leq \zeta$. Then, applying union bound, we know that, with probability at most $|S|\zeta = \varepsilon$, $s$ is not good for all eigenvalues in $S$. The claim in our theorem thus follows. $\qquad\square$

The term "consistent" means that, with high probability, the algorithm can choose a good shift, and thus the phase estimation result does not depend on the measurement. This is crucial for the later construction of the oracle for quantum maxima finding.

**Theorem A.3.** *Let $U$ be an $n \times n$ unitary matrix. Suppose that:*

$$U|0\rangle|0\rangle = \sqrt{p}|0\rangle|\phi_0\rangle + \sqrt{1-p}|1\rangle|\phi_1\rangle,$$

*where $p \in (0,1)$, $|\phi_0\rangle$ and $|\phi_1\rangle$ are normalized pure quantum states. Then there exists a quantum algorithm such that, for every $\varepsilon > 0$ and $\delta > 0$, on input $O(\log(\varepsilon^{-1}))$-bit random string, with probability at least $1 - \varepsilon$, the algorithm outputs $f(s,p)$ such that*

$$|f(s,p) - p| \leq \delta,$$

*using $O(\epsilon^{-4}\delta^{-1})$ queries to $U$ and $O(\varepsilon^{-4}\delta^{-1}\log(n))$ one- and two-qubit quantum gates.*

*Proof.* Denote

$$Q = -U(I - 2|0\rangle\langle 0| \otimes |0\rangle\langle 0|)U^\dagger(I - 2|0\rangle\langle 0| \otimes I).$$

Then we can compute

$$Q|0\rangle|\phi_0\rangle = (1-2p)|0\rangle|\phi_0\rangle - 2\sqrt{p(1-p)}|1\rangle|\phi_1\rangle,$$

and

$$Q|1\rangle|\phi_1\rangle = 2\sqrt{p(1-p)}|0\rangle|\phi_0\rangle + (1-2p)|1\rangle|\phi_1\rangle.$$

To simplify the above equations, let $\theta_p$ be the unique number in $(0, \pi/2)$ such that

$$\sin^2(\theta_p) = p,$$

then $Q$ has two eigenvectors in the following form:

$$|\psi_\pm\rangle = \frac{1}{\sqrt{2}}(|0\rangle|\phi_0\rangle \pm \mathrm{i}|1\rangle|\phi_1\rangle),$$

such that

$$Q|\psi_\pm\rangle = \exp(\pm\mathrm{i}2\theta_p)|\psi_\pm\rangle.$$

Now, notice that:

$$U|0\rangle|0\rangle = \frac{-\mathrm{i}}{\sqrt{2}}(\exp(\mathrm{i}\theta_p)|\psi_+\rangle - \exp(-\mathrm{i}\theta_p)|\psi_-\rangle),$$

we can apply our Algorithm 4 with oracle query to $Q$, precision parameter being $\delta' = \delta/\pi$, error parameter being $\varepsilon'$ ($\varepsilon'$ is a fixed parameter that will be decided later), the input state being $U|0\rangle|0\rangle$, and a $\log(4/\varepsilon')$-bit random string $s$.

Using Theorem A.2, with probability $1 - \varepsilon'$, we can obtain a state that is $O(\sqrt{\varepsilon'})$-close to

$$\frac{-\mathrm{i}}{\sqrt{2}}\left(\exp(\mathrm{i}\theta_p)|\psi_+\rangle \otimes \left|f\left(s, \frac{\theta_p}{\pi}\right)\right\rangle - \exp(-\mathrm{i}\theta_p)|\psi_-\rangle \otimes \left|f\left(s, 1 - \frac{\theta_p}{\pi}\right)\right\rangle\right)$$

in trace distance. Thus, measuring the last register, with probability at least $1 - \sqrt{\varepsilon'}$ we will get the result $\gamma$, which is either $f(s, \theta_p/\pi)$ or $f(s, 1 - \theta_p/\pi)$. Then output $\sin^2(\gamma\pi)$ as the estimate of $p$. Notice that

$$\sin^2(\theta_p) = \sin^2(\pi - \theta_p) = p,$$

and $\sin^2(\cdot)$ is 2-Lipschitz and even, we know the additive error is bounded by

$$|\sin^2(\gamma\pi) - p| \leq |\pi\gamma - \theta_p| \leq \pi\delta' = \delta.$$

Thus, by choosing $\varepsilon' = \Theta(\varepsilon^2)$ we can have $(1 - \varepsilon')(1 - \sqrt{\varepsilon'}) \geq 1 - \varepsilon$, and thus the result follows. $\qquad\square$

Then we state and prove the error-reduced version of consistent amplitude estimation.

**Theorem A.4** (Restatement of Theorem 2.3). *Let $U$ be an $n \times n$ unitary matrix. Suppose that:*

$$U|0\rangle|0\rangle = \sqrt{p}|0\rangle|\phi_0\rangle + |1\rangle|\phi_1\rangle,$$

*where $p \in (0,1)$, $|\phi_0\rangle$ and $|\phi_1\rangle$ are normalized pure quantum states. Then there exists a quantum algorithm such that, for every $\varepsilon > 0$ and $\delta > 0$, on input an $O(r)$-bit random string $s$, with probability at least $1 - \exp(-\Omega(r))$, the algorithm outputs $f(s,p)$ such that*

$$|f(s,p) - p| \leq \delta,$$

*using $O(r\delta^{-1})$ queries to $U$ and $O(r\delta^{-1}\log(n))$ one- and two-qubit quantum gates.*

*Proof.* Split the input random string $s$ into $r$ strings $s_1, s_2, \ldots, s_r$ of length $\Theta(1)$. For each $i \in [r]$, we use the algorithm described in the Theorem A.3 with input string $s_i$ and error parameter $\varepsilon = 1/10$. Thus, we have, for each $i \in [r]$,

$$\mathbb{P}(|f(s_i,p) - p| \leq \delta) \geq \frac{9}{10}.$$

Now we set $f^*(s,p)$ to be the median of the estimations $f(s_i,p)$ for $i \in [r]$ to be the output estimation. We claim that, with high probability it will be the desired estimation. To show that, we define random variables $X_i$ for $i \in [r]$ as follows:

$$X_i = \begin{cases} 1, & \text{if } |f(s_i,p) - p| \leq \delta, \\ 0, & \text{otherwise.} \end{cases}$$

Noticing that $\mathbb{E}[\sum_{i=1}^r X_i] \geq 9r/10$, by Chernoff bound, we have:

$$\mathbb{P}\left(\sum_{i=1}^r X_i < \frac{r}{2}\right) \leq \exp\left(-\frac{8r}{45}\right).$$

Thus with probability at least $1 - \exp(-8r/45)$, we know that at least half of the estimations fall into the interval $[p-\delta, p+\delta]$, and then the median $f^*(s,p)$ must return a result in this interval. $\square$

## B    PROOF OF THE PRE-PROCESSING ALGORITHM

*Proof of Theorem 3.1.* Since $V$ is a $\beta$-amplitude encoding of the vector $u \in \mathbb{R}^n_{\geq 0}$, we know that

$$(\mathsf{XOR}_{D,C})^\dagger (V \otimes I_D)(\mathsf{XOR}_{D,C})|i\rangle_D|0\rangle_C|0\rangle_A|0\rangle_B = |i\rangle_D \otimes \left(\sqrt{\frac{u_i}{\beta}}|0\rangle_C|0\rangle_A|\psi_i\rangle_B + \sqrt{1 - \frac{u_i}{\beta}}|1\rangle_C|g\rangle_{AB}\right),$$

where $|g\rangle_{AB}$ is a normalized pure state, and $\mathsf{XOR}_{D,C}$ stands for the following unitary gate:

$$\mathsf{XOR}_{D,C}|i\rangle_D|j\rangle_C = |i\rangle_D|i \oplus j\rangle_C.$$

Thus, by applying Theorem 2.3 to the unitary $(\mathsf{XOR}_{D,C})^\dagger (V \otimes I_D)(\mathsf{XOR}_{D,C})$, with $\delta = 1/2\beta$ and $r$ to be decided later, we know that there exists a quantum algorithm such that, on input an $O(r)$-bit random string $s$ and the state $|i\rangle_D|0\rangle_{ABC}$, the algorithm outputs $f(s, u_i/\beta)$ with probability at least $1 - \exp(-\Omega(r))$ such that

$$\left|f\left(s, \frac{u_i}{\beta}\right) - \frac{u_i}{\beta}\right| \leq \frac{1}{2\beta},$$

using $O(r\delta^{-1}) = O(r\beta)$ queries to $V$ and $O(r\beta\log(n))$ one- and two-qubit gates.

Thus, by setting

$$\widetilde{u}_i = \beta f\left(s, \frac{u_i}{\beta}\right) + \frac{1}{2},$$

we know $u_i \leq \widetilde{u}_i \leq u_i + 1$. Thus, by storing $\widetilde{u}_i$ into a new quantum register, we can obtain a unitary $U_{\widetilde{u}}$ that is $\exp(-\Omega(r))$-close in operator norm to the following unitary $\mathcal{O}_{\widetilde{u}}$:

$$\mathcal{O}_{\widetilde{u}}|i\rangle|0\rangle = |i\rangle|\widetilde{u}_i\rangle,$$

and $U_{\widetilde{u}}$ uses $O(r\delta^{-1}) = O(r\beta)$ queries to $V$ and $O(r\beta\log(n))$ one- and two-qubit gates.

Then, Theorem 2.1 implies that with probability $2/3 - \sqrt{n\ell}\exp(-\Omega(r))$, $\mathsf{FindMax}(U_{\widetilde{u}}, n, \ell)$ finds a set $S$ containing the indices of the largest $\ell$ elements of $\widetilde{u}_i$ using $O(\sqrt{n\ell})$ queries to $U_{\widetilde{u}}$. Thus, by choosing $r = \Theta(\log(n\ell/\varepsilon))$ and a fixed $O(r)$-bit random string $s$, the probability that the algorithm outputs the correct set is $\Omega(1)$. Finally, by repeating $\mathsf{FindMax}(U_{\widetilde{u}}, n, \ell)$ for $\Theta(\log(1/\varepsilon))$ times (using the same random string $s$), we can output the correct set $S$ with probability at least $1 - \varepsilon/2$. The above procedure uses in total $O(\beta\sqrt{n\ell}\log(n\ell/\varepsilon)\log(1/\varepsilon))$ queries to $V$.

After successfully getting $S$, we again use Theorem 2.3, with the same $r = \Theta(\log(n\ell/\varepsilon))$, the same string $s$ and $\delta = 1/2\beta$, to obtain and store $\widetilde{u}_i$ classically for all $i \in S$ with success probability at least $1 - \varepsilon/2$, using $O(\ell\beta\log(n\ell/\varepsilon))$ queries to $V$. Noting that $0 < \ell \leq n$, we get the desired theorem. $\qquad\square$

## C  PROOF OF THE MAIN THEOREM

In this part, we give a proof for Theorem 3.2. We first notice the following propositions.

**Proposition C.1.** *Let $S \subseteq [n]$ be a set with cardinality $\ell$. Suppose that there is an instance $\mathcal{T}_{\exp}$ of $\mathsf{SamplerTree}$ specified in Lemma 2.9 which maintains an $n$-dimensional vector $\widetilde{u}^{\exp} \in \mathbb{R}_{\geq 0}^n$, where $\widetilde{u}_i^{\exp} = \exp(\widetilde{u}_i)/W$ for $i \in S$, and $\widetilde{u}_i^{\exp} = \exp(\widetilde{u}_{\min})/W$ for $i \notin S$, with $W = (n-\ell)\exp(\widetilde{u}_{\min}) + \sum_{i\in S}\exp(\widetilde{u}_i)$. Then for $\varepsilon_1 \in (0, 1/2)$ we can implement the unitary $U_{\text{guess}}$ which satisfies*

$$\left\| U_{\text{guess}}|0\rangle - \left( \sum_{i\in S}\sqrt{\frac{\exp(\widetilde{u}_i)}{W}}|i\rangle + \sum_{i\notin S}\sqrt{\frac{\exp(\widetilde{u}_{\min})}{W}}|i\rangle \right) \right\| \leq \varepsilon_1,$$

*in $O(\log^2(n)\log^{5/2}(n/\varepsilon_1))$ time.*

*Proof.* This is direct by the state preparation operation specified in Lemma 2.9. $\qquad\square$

**Lemma C.2.** *Let $S \subseteq [n]$ be a set with cardinality $\ell$. Let $V$ be a $(\beta, \lceil\log_2(n)\rceil, O(1), O(1))$-amplitude-encoding of $u \in \mathbb{R}_{\geq 0}^n$. Suppose that there is an instance $\mathcal{T}$ of $\mathsf{SamplerTree}$ specified in Lemma 2.9 which maintains an $n$-dimensional vector $\check{u} \in \mathbb{R}_{\geq 0}^n$, where $\check{u}_i = \widetilde{u}_i$ for $i \in S$, and $\check{u}_i = \widetilde{u}_{\min}$ for $i \notin S$, with $\widetilde{u}_i$ and $\widetilde{u}_{\min}$ defined in Theorem 3.1. Define $w_i = (\check{u}_i - u_i)/(2\beta) - 1$. Then for $\varepsilon_q \in (0, 1/2)$, $t = O(\beta + \log(1/\varepsilon_q))$, and $d = O(\sqrt{t\log(1/\varepsilon_q)})$, we can implement $U_q$ that is a $(1, O(1), \varepsilon_q)$-block-encoding of $\mathrm{diag}(q_{\beta,t,d}(w)/4)$ where $q_{\beta,t,d}(\cdot)$ is the polynomial defined in Theorem 2.8, using $O(\sqrt{t\log(1/\varepsilon_q)})$ queries to $V$, in $O(\sqrt{t\log(1/\varepsilon_q)}(\log(n) + \log^{5/2}(\beta t/\varepsilon_q)))$ time.*

*Proof.* From the assumption and Lemma 2.9, we know that using the query operation for $\mathcal{T}$ of $\mathsf{SamplerTree}$, we can implement we can implement $U_{\check{u}}$ that is a $(1, O(1), \varepsilon_2)$-block-encoding of $\mathrm{diag}(\check{u}/\beta)$ in $O(\log(n) + \log^{5/2}(\beta/\varepsilon_2))$ time, for some $\varepsilon_2 \in (0, 1/2)$.

From our assumption that $V$ is a $(\beta, \lceil\log_2(n)\rceil, O(1), O(1))$-amplitude-encoding of $u$, using Lemma 2.10, we can implement $U_u$ that is a $(\beta, O(1), 0)$-block-encoding of $\mathrm{diag}(u)$, using $O(1)$ queries to $V$. Note that $U_u$ is a $(1, O(1), 0)$-block-encoding of $\mathrm{diag}(u/\beta)$.

By the technique of linear combination of unitaries (Lemma 2.5), with the $(2\sqrt{6}, 2, 0)$-state-preparation-pair $(P_L, P_R)$ for the vector $(1, -1, -2)$ satisfying

$$P_L|0\rangle = \frac{1}{\sqrt{6}}|0\rangle - \frac{1}{\sqrt{6}}|1\rangle - \frac{\sqrt{2}}{\sqrt{3}}|2\rangle,$$

$$P_R|0\rangle = \frac{1}{2}|0\rangle + \frac{1}{2}|1\rangle + \frac{1}{2}|2\rangle + \frac{1}{2}|3\rangle,$$

we can implement a $(2\sqrt{6}, O(1), 2\sqrt{6}\varepsilon_2)$-block-encoding $U_{\mathsf{LCU}}$ of

$$\mathrm{diag}(\check{u}/\beta) - \mathrm{diag}(u/\beta) - 2I = 2\,\mathrm{diag}(w),$$

using $O(1)$ queries to $U_{\check{u}}$ and $U_u$. Note that $U_{\mathsf{LCU}}$ is a $(\sqrt{6}, O(1), 2\sqrt{6}\varepsilon_2)$-block-encoding of $\mathrm{diag}(w)$. Using Lemma 2.4, we can implement a $(1, O(1), O(\varepsilon_2))$-block-encoding $U_w$ of $\mathrm{diag}(w)$, using $O(1)$ queries to $U_{\mathsf{LCU}}$.

To summarize the above discussions, $U_w$ can be constructed using $O(1)$ queries to $V$ and $O(\log(n) + \log^{5/2}(\beta/\varepsilon_2))$ time.

Now, noting that by the choice of $t$ and $d$, by Theorem 2.8, we have:

$$\sup_{x \in [-1,1]} |q_{\beta,t,d}(x) - \exp(-\beta - \beta x)| \le \varepsilon_q.$$

Thus, for $\varepsilon_q \in (0, 1/2)$,

$$\sup_{x \in [-1,1]} |q_{\beta,t,d}(x)| \le \sup_{x \in [-1,1]} |\exp(-\beta - \beta x)| + \varepsilon_q \le 1 + \varepsilon_q < 2.$$

Therefore, the polynomial $q_{\beta,t,d}(\cdot)/4$ satisfies the requirement of Theorem 2.6. By using Theorem 2.6 with $\delta = \varepsilon_q/2$, we can implement a $(1, O(1), \varepsilon_q/2 + O(d\sqrt{\varepsilon_2}))$-block-encoding of $\mathrm{diag}(q_{\beta,t,d}(w)/4)$, using $O(d)$ queries to $\mathrm{diag}(w)$ and in $O(d)$ time. Thus by choosing $\varepsilon_2 = \Theta(\varepsilon_q^2/d^2)$, we can implement $U_{\mathsf{ET}}$ using $O(\sqrt{t \log(1/\varepsilon_q)})$ queries to $V$, in $O(\sqrt{t \log(1/\varepsilon_q)}(\log(n) + \log^{5/2}(\beta t/\varepsilon_q)))$ time. $\qquad\square$

Using similar techniques in Hamoudi (2022); Gao et al. (2023), we can prove the following lemmas:

**Lemma C.3.** *For vector $u \in \mathbb{R}_{\ge 0}^n$, $\beta > 0$ that satisfies $\beta \ge \|u\|_1$. Let $W, S, \widetilde{u}_i$ and $\widetilde{u}_{\min}$ be the same as defined in Algorithm 2, then for $t = O(\beta + \log(\varepsilon_q^{-1}))$ and $d = O(\sqrt{\beta \log(\varepsilon_q^{-1})})$, where $\varepsilon_q \in (0, 1/2)$ is an approximation parameter that will be decided later, let*

$$|u_{\mathrm{post}}\rangle = \frac{1}{4}\left( \sum_{i \in S} q_{\beta,t,d}\left(\frac{\widetilde{u}_i - u_i}{2\beta} - 1\right)\sqrt{\frac{\exp(\widetilde{u}_i)}{W}}|i\rangle + \sum_{i \notin S} q_{\beta,t,d}\left(\frac{\widetilde{u}_{\min} - u_i}{2\beta} - 1\right)\sqrt{\frac{\exp(\widetilde{u}_{\min})}{W}}|i\rangle \right),$$

*and $E = \sum_{j=1}^n \exp(u_i)$. Then*

$$\frac{\ell}{16\mathrm{e}n} - \frac{\varepsilon_q}{8} \le \frac{E}{16W} - \frac{\varepsilon_q}{8} \le \|\,|u_{\mathrm{post}}\rangle\|^2 \le \frac{E}{16W} + \frac{3\varepsilon_q}{16}.$$

*Proof.* Notice that $u_i \le \widetilde{u}_i \le u_i + 1$, we have:

$$W = (n - \ell)\exp(\widetilde{u}_{\min}) + \sum_{i \in S} \exp(\widetilde{u}_i) \le \mathrm{e}\left( (n - \ell)\exp(u_{\min}) + \sum_{i \in S} \exp(u_i) \right).$$

Note that:

$$\frac{(n - \ell)\exp(u_{\min}) + \sum_{i \in S} \exp(u_i)}{E} \le \frac{n - \ell}{\ell} + 1 = \frac{n}{\ell}.$$

Thus, combining the above two inequalities, we have:

$$\frac{E}{W} \ge \frac{\ell}{n\mathrm{e}}.$$

Now, by our choice of $t$ and $d$, using Theorem 2.8, we know

$$\sup_{x \in [-1,1]} |q_{\beta,t,d}(x) - \exp(-\beta - \beta x)| \le \varepsilon_q.$$

Therefore, we have, for $i \in S$:

$$\left| q_{\beta,t,d}\left(\frac{\widetilde{u}_i - u_i}{2\beta} - 1\right) - \exp\left(\frac{u_i - \widetilde{u}_i}{2}\right) \right| \le \varepsilon_q;$$

and for $i \notin S$:

$$\left| q_{\beta,t,d}\left(\frac{\widetilde{u}_{\min} - u_i}{2\beta} - 1\right) - \exp\left(\frac{u_i - \widetilde{u}_{\min}}{2}\right) \right| \le \varepsilon_q.$$

We have

$$\left( q_{\beta,t,d}\left(\frac{\widetilde{u}_i - u_i}{2\beta} - 1\right) \right)^2 \ge \exp(u_i - \widetilde{u}_i) - 2\varepsilon_q,$$

for all $i \in S$ as $u_i \le \widetilde{u}_i$. Similarly, for all $i \notin S$, we have

$$\left( q_{\beta,t,d}\left(\frac{\widetilde{u}_{\min} - u_i}{2\beta} - 1\right) \right)^2 \ge \exp(u_i - \widetilde{u}_{\min}) - 2\varepsilon_q.$$

Thus we can deduce

$$
\begin{aligned}
\||u_{\text{post}}\rangle\|^2 &= \frac{1}{16}\left(\sum_{i\in S}\left(q_{\beta,t,d}\left(\frac{\widetilde{u}_i - u_i}{2\beta} - 1\right)\right)^2 \frac{\exp(\widetilde{u}_i)}{W} + \sum_{i\notin S}\left(q_{\beta,t,d}\left(\frac{\widetilde{u}_{\min} - u_i}{2\beta} - 1\right)\right)^2 \frac{\exp(\widetilde{u}_{\min})}{W}\right) \\
&\geq \frac{1}{16}\left(\sum_{i\in S}(\exp(u_i - \widetilde{u}_i) - 2\varepsilon_q)\frac{\exp(\widetilde{u}_i)}{W} + \sum_{i\notin S}(\exp(u_i - \widetilde{u}_{\min}) - 2\varepsilon_q)\frac{\exp(\widetilde{u}_{\min})}{W}\right) \\
&= \frac{E}{16W} - \frac{\varepsilon_q}{8} \\
&\geq \frac{\ell}{16\mathrm{e}n} - \frac{\varepsilon_q}{8}.
\end{aligned}
$$

For the other side, note that for all $i \in S$, we have:

$$
\left(q_{\beta,t,d}\left(\frac{\widetilde{u}_i - u_i}{2\beta} - 1\right)\right)^2 \leq \exp(u_i - \widetilde{u}_i) + 3\varepsilon_q,
$$

and for all $i \notin S$, we have:

$$
\left(q_{\beta,t,d}\left(\frac{\widetilde{u}_{\min} - u_i}{2\beta} - 1\right)\right)^2 \leq \exp(u_i - \widetilde{u}_{\min}) + 3\varepsilon_q.
$$

Thus we have:

$$
\begin{aligned}
\||u_{\text{post}}\rangle\|^2 &= \frac{1}{16}\left(\sum_{i\in S}\left(q_{\beta,t,d}\left(\frac{\widetilde{u}_i - u_i}{2\beta} - 1\right)\right)^2 \frac{\exp(\widetilde{u}_i)}{W} + \sum_{i\notin S}\left(q_{\beta,t,d}\left(\frac{\widetilde{u}_{\min} - u_i}{2\beta} - 1\right)\right)^2 \frac{\exp(\widetilde{u}_{\min})}{W}\right) \\
&\leq \frac{1}{16}\left(\sum_{i\in S}(\exp(u_i - \widetilde{u}_i) + 3\varepsilon_q)\frac{\exp(\widetilde{u}_i)}{W} + \sum_{i\notin S}(\exp(u_i - \widetilde{u}_{\min}) + 3\varepsilon_q)\frac{\exp(\widetilde{u}_{\min})}{W}\right) \\
&= \frac{E}{16W} + \frac{3\varepsilon_q}{16}.
\end{aligned}
$$

$\square$

**Lemma C.4.** *Let*

$$
|u_{\text{post}}\rangle = \frac{1}{4}\left(\sum_{i\in S}q_{\beta,t,d}\left(\frac{\widetilde{u}_i - u_i}{2\beta} - 1\right)\sqrt{\frac{\exp(\widetilde{u}_i)}{W}}|i\rangle + \sum_{i\notin S}q_{\beta,t,d}\left(\frac{\widetilde{u}_{\min} - u_i}{2\beta} - 1\right)\sqrt{\frac{\exp(\widetilde{u}_{\min})}{W}}|i\rangle\right),
$$

*and*

$$
|u_{\text{Gibbs}}\rangle = \sum_{i=1}^{n}\sqrt{\frac{\exp(u_i)}{E}}|i\rangle,
$$

*where $E = \sum_{j=1}^{n}\exp(u_i)$.*

*Then the trace disance between the states $|u_{\text{Gibbs}}\rangle$ and $|u_{\text{post}}\rangle/\||u_{\text{post}}\rangle\|$ is no more than*

$$
\sqrt{\frac{15\varepsilon_q n}{\ell}}.
$$

*Proof.* Notice that

$$
\begin{aligned}
\langle u_{\text{post}} | u_{\text{Gibbs}} \rangle &= \frac{1}{4} \left( \sum_{i \in S} q_{\beta,t,d}(w_i) \sqrt{\frac{\exp(\widetilde{u}_i)}{W}} \sqrt{\frac{\exp(u_i)}{E}} + \sum_{i \notin S} q_{\beta,t,d}(w_i) \sqrt{\frac{\exp(\widetilde{u}_{\min})}{W}} \sqrt{\frac{\exp(u_i)}{E}} \right) \\
&\geq \frac{1}{4} \left( \sum_{i \in S} \left( \exp\left( \frac{u_i - \widetilde{u}_i}{2} \right) - \varepsilon_q \right) \sqrt{\frac{\exp(\widetilde{u}_i)}{W}} \sqrt{\frac{\exp(u_i)}{E}} \right) \\
&\quad + \frac{1}{4} \left( \sum_{i \notin S} \left( \exp\left( \frac{u_i - \widetilde{u}_{\min}}{2} \right) - \varepsilon_q \right) \sqrt{\frac{\exp(\widetilde{u}_{\min})}{W}} \sqrt{\frac{\exp(u_i)}{E}} \right) \\
&= \frac{1}{4} \left( \frac{\sum_{i=1}^n \exp(u_i)}{\sqrt{WE}} - \varepsilon_q \left( \sum_{i \in S} \sqrt{\frac{\exp(\widetilde{u}_i)}{W}} \sqrt{\frac{\exp(u_i)}{E}} + \sum_{i \notin S} \sqrt{\frac{\exp(\widetilde{u}_{\min})}{W}} \sqrt{\frac{\exp(u_i)}{E}} \right) \right) \\
&\geq \sqrt{\frac{E}{16W}} - \frac{\varepsilon_q}{4}.
\end{aligned}
$$

The last step is by using Cauchy's inequality to bound the coefficient of $\varepsilon_q$.

Using Lemma C.3, we have:

$$
\| |u_{\text{post}}\rangle \|^2 \leq \frac{E}{16W} + \frac{3\varepsilon_q}{16}.
$$

This means

$$
\frac{1}{\| |u_{\text{post}}\rangle \|^2} \geq \frac{16}{\frac{E}{W} + 3\varepsilon_q} \geq \frac{16W}{E} - \frac{48\varepsilon_q W^2}{E^2},
$$

for $\varepsilon_q > 0$.

Noting that, by definition, we have $E \leq W$. Thus, we have:

$$
\begin{aligned}
\frac{|\langle u_{\text{post}} | u_{\text{Gibbs}} \rangle|^2}{\| |u_{\text{post}}\rangle \|^2} &\geq \frac{1}{\| |u_{\text{post}}\rangle \|^2} \left( \frac{E}{16W} - \frac{\varepsilon_q}{8} \right) \\
&\geq 1 - \frac{5\varepsilon_q W}{E} \\
&\geq 1 - \frac{15\varepsilon_q n}{\ell}.
\end{aligned}
$$

Thus, the trace disance between the states $|u_{\text{Gibbs}}\rangle$ and $|u_{\text{post}}\rangle / \| |u_{\text{post}}\rangle \|$ is no more than

$$
\sqrt{\frac{15\varepsilon_q n}{\ell}}.
$$

$\square$

We need the following lemma for vector norms to bound error.

**Lemma C.5.** *For two non-zero vectors $|u\rangle$ and $|v\rangle$ with $\| |u\rangle - |v\rangle \| \leq \varepsilon$ for some $\varepsilon \geq 0$, then*

$$
\left\| \frac{|u\rangle}{\| |u\rangle \|} - \frac{|v\rangle}{\| |v\rangle \|} \right\| \leq \frac{2\varepsilon}{\| |u\rangle \|}.
$$

*Proof.*

$$
\begin{aligned}
\left\| \frac{|u\rangle}{\| |u\rangle \|} - \frac{|v\rangle}{\| |v\rangle \|} \right\| &\leq \left\| \frac{|u\rangle}{\| |u\rangle \|} - \frac{|v\rangle}{\| |u\rangle \|} \right\| + \left\| \frac{|v\rangle}{\| |u\rangle \|} - \frac{|v\rangle}{\| |v\rangle \|} \right\| \\
&= \frac{1}{\| |u\rangle \|} \| |u\rangle - |v\rangle \| + \| |v\rangle \| \left| \frac{1}{\| |u\rangle \|} - \frac{1}{\| |v\rangle \|} \right| \\
&\leq \frac{\varepsilon}{\| |u\rangle \|} + \frac{\varepsilon}{\| |u\rangle \|} = \frac{2\varepsilon}{\| |u\rangle \|}.
\end{aligned}
$$

$\square$

We are now ready to prove Theorem 3.2.

*Proof of Theorem 3.2.* Using Proposition C.1, we know that after initialization and assignment of the instance $\mathcal{T}_{\text{exp}}$ of SamplerTree in $O(\ell \log(n))$ time, we can implement $U_{\text{guess}}$ in $O(\log^2(n) \log^{5/2}(n/\varepsilon))$ time. Then using Lemma C.2 with $\varepsilon_q = \ell\varepsilon^2/120n$, we know we can implement $U_{\text{ET}}$ using $O(\sqrt{\beta \log(n/\ell\varepsilon)} + \log(n/\ell\varepsilon))$ queries to $V$, in $O((\sqrt{\beta \log(n/\ell\varepsilon)} + \log(n/\ell\varepsilon)) \log^{5/2}(n\beta/\ell\varepsilon))$ time. In addition, we know

$$U_{\text{ET}}(I_a \otimes U_{\text{guess}})|0\rangle|0\rangle = |0\rangle|\hat{u}_{\text{post}}\rangle + |1\rangle|g\rangle,$$

where

$$\||\hat{u}_{\text{post}}\rangle - |u_{\text{post}}\rangle\| \le 2\varepsilon_q,$$

and $|g\rangle$ is an unnormalized garbage state.

In Lemma C.3, it is proved that

$$\||u_{\text{post}}\rangle\| \ge \sqrt{\frac{\ell}{16en} - \frac{\varepsilon_q}{8}} = \Omega\left(\sqrt{\frac{\ell}{n}}\right).$$

Thus, we can deduce that:

$$\||\hat{u}_{\text{post}}\rangle\| \ge \||u_{\text{post}}\rangle\| - 2\varepsilon_q = \Omega\left(\sqrt{\frac{\ell}{n}}\right).$$

Therefore, by applying Theorem 2.2, with probability at least $1 - \varepsilon/k$, we can get the normalized state

$$\frac{|\hat{u}_{\text{post}}\rangle}{\||\hat{u}_{\text{post}}\rangle\|}$$

using $O(\sqrt{n/\ell} \log(k/\varepsilon))$ queries to $U_{\text{ET}}$ and $U_{\text{guess}}$. To summarize, the above procedures uses $O(\sqrt{n/\ell} \log(k/\varepsilon)(\sqrt{\beta \log(n/\ell\varepsilon)} + \log(n/\ell\varepsilon)))$ queries to $V$, and in $O(\sqrt{n/\ell} \log(k/\varepsilon)((\sqrt{\beta \log(n/\ell\varepsilon)} + \log(n/\ell\varepsilon)) \log^{5/2}(n\beta/\ell\varepsilon) + \log^2(n) \log^{5/2}(n/\varepsilon)))$ time.

By Lemma C.5, we have:

$$\left\| \frac{|u_{\text{post}}\rangle}{\||u_{\text{post}}\rangle\|} - \frac{|\hat{u}_{\text{post}}\rangle}{\||\hat{u}_{\text{post}}\rangle\|} \right\| \le \frac{4\varepsilon_q}{\||u_{\text{post}}\rangle\|} \le \frac{4\varepsilon_q}{\sqrt{\ell/(16en) - \varepsilon_q/8}}.$$

Using Lemma C.4, we know that for the state defined as follows:

$$|u_{\text{Gibbs}}\rangle = \sum_{i=1}^{n} \sqrt{\frac{\exp(u_i)}{E}}|i\rangle,$$

where $E = \sum_{j=1}^{n} \exp(u_i)$, we have:

$$\frac{1}{2}\left\| |u_{\text{Gibbs}}\rangle\langle u_{\text{Gibbs}}| - \frac{|u_{\text{post}}\rangle\langle u_{\text{post}}|}{\||u_{\text{post}}\rangle\|^2} \right\|_1 \le \sqrt{\frac{15\varepsilon_q n}{\ell}}.$$

Note that for any two normalized pure states $|\phi\rangle$ and $|\psi\rangle$, we have:

$$\frac{1}{2}\||\psi\rangle\langle\psi| - |\phi\rangle\langle\phi|\|_1 \le \||\psi\rangle - |\phi\rangle\|.$$

Therefore, we have

$$\frac{1}{2}\left\| |u_{\text{Gibbs}}\rangle\langle u_{\text{Gibbs}}| - \frac{|\hat{u}_{\text{post}}\rangle\langle\hat{u}_{\text{post}}|}{\||\hat{u}_{\text{post}}\rangle\|^2} \right\|_1$$

$$\le \frac{1}{2}\left\| |u_{\text{Gibbs}}\rangle\langle u_{\text{Gibbs}}| - \frac{|u_{\text{post}}\rangle\langle u_{\text{post}}|}{\||u_{\text{post}}\rangle\|^2} \right\|_1 + \frac{1}{2}\left\| \frac{|u_{\text{post}}\rangle\langle u_{\text{post}}|}{\||u_{\text{post}}\rangle\|^2} - \frac{|\hat{u}_{\text{post}}\rangle\langle\hat{u}_{\text{post}}|}{\||\hat{u}_{\text{post}}\rangle\|^2} \right\|_1$$

$$\le \sqrt{\frac{15\varepsilon_q n}{\ell}} + \left\| \frac{|u_{\text{post}}\rangle}{\||u_{\text{post}}\rangle\|} - \frac{|\hat{u}_{\text{post}}\rangle}{\||\hat{u}_{\text{post}}\rangle\|} \right\|$$

$$\le \sqrt{\frac{15\varepsilon_q n}{\ell}} + \frac{4\varepsilon_q}{\sqrt{\ell/(16en) - \varepsilon_q/8}} \le \varepsilon.$$

The last inequality is by our choice of $\varepsilon_q = \ell\varepsilon^2/120n$.

Thus by measuring the state

$$\frac{|\hat{u}_{\text{post}}\rangle}{\||\hat{u}_{\text{post}}\rangle\|},$$

we will get a sample that is from a distribution which is $\varepsilon$-close to $\text{Gibbs}(u)$ in total variation distance. This can be done by amplitude amplification to boost the success probability to $1 - \varepsilon/2k$.

After repeating the state preparation and measurement procedure $k$ times, the algorithm will output all the measurement results. The cost except the pre-processing step is

$$O\Big(\ell\log(n) + k\sqrt{n/\ell}\log(k/\varepsilon)\Big(\big(\sqrt{\beta\log(n/\ell\varepsilon)} + \log(n/\ell\varepsilon)\big)\log^{5/2}(n\beta/\ell\varepsilon) + \log^2(n)\log^{5/2}(n/\varepsilon)\Big)\Big)$$

time, and using

$$O\Big(k\sqrt{n/\ell}\big(\sqrt{\beta\log(n/\ell\varepsilon)} + \log(n/\ell\varepsilon)\big)\log(k/\varepsilon)\Big)$$

queries to $V$.

Therefore, by setting:

$$\ell = \frac{k\log(k/\varepsilon)}{\beta^{1/2}\log^{1/2}(n/\varepsilon)\log(1/\varepsilon)},$$

we can achieve the best query complexity of $V$ being

$$O\Big(\sqrt{nk}\big(\beta^{3/4} + \beta^{1/4}\log^{1/2}(n/\varepsilon)\big)\log^{1/2}(1/\varepsilon)\log^{3/4}(n/\varepsilon)\log^{1/2}(k/\varepsilon)\Big).$$

$\square$

## D  FROM MATRIX ZERO-SUM GAMES TO LINEAR PROGRAMMING

In this part, we explain reducing solving linear programming to solving matrix zero-sum games. The outline of reduction follows van Apeldoorn & Gilyén (2019b). We assume that the linear programming problem is formulated as the following standard form:

$$\begin{aligned}
\underset{x \in \mathbb{R}^n}{\text{minimize}} \quad & c^\mathsf{T} x \\
\text{subject to} \quad & \mathbf{A}x \le b, \\
& x \ge 0
\end{aligned} \tag{4}$$

The dual of the above program can be written as follows:

$$\begin{aligned}
\underset{y \in \mathbb{R}^m}{\text{maximize}} \quad & b^\mathsf{T} y \\
\text{subject to} \quad & \mathbf{A}^\mathsf{T} y \le c, \\
& y \ge 0
\end{aligned} \tag{5}$$

Let $R$ be a prior bound on the $\ell_1$-norm of the optimal dual solution and $r$ be a prior constant bound on the $\ell_1$-norm of the optimal primal solution. Consider the matrix $\mathbf{A}'$ defined as follows:

$$\mathbf{A}' = \begin{pmatrix} \mathbf{1} & 1 & -1 \\ -\mathbf{1} & 1 & 1 \\ -b^\mathsf{T} & 0 & \alpha/R \\ \mathbf{A}^\mathsf{T} & 0 & -c/R \end{pmatrix} \tag{6}$$

By the strong duality theorem, the optimal values of the above primal (Equation (2)) and dual programs (Equation (5)) are equal. We have the following theorem for the reduction:

**Theorem D.1** ((van Apeldoorn & Gilyén, 2019b, Lemma 12)). *Determining the $\varepsilon' = \varepsilon/(6R(r+1))$-approximate value of the game $\mathbf{A}'$ suffices to decide whether the optimal value $\text{Opt}$ for the linear programming problem defined in Equation (2) satisfies $\text{Opt} \ge \alpha$ or $\text{Opt} \le \alpha - \varepsilon$.*

Notice that in Corollary 4.3, the quantum algorithm only returns an $\varepsilon$-approximate optimal strategy pair of the game, rather than the optimal value. Thus, we need the following proposition to estimate the value of the matrix game. For completeness, we first recall Hoeffding's inequality here.

**Theorem D.2** (Hoeffding's inequality, Hoeffding (1963)). *Let $0 \leq X_i \leq 1$ be independent random variables for every $1 \leq i \leq n$. Let $X = \sum_{i=1}^{n} X_i/n$ and $\varepsilon > 0$. Then,*

$$\Pr[X \leq \mathbb{E}[X] - \varepsilon] \leq \exp(-2n\varepsilon^2).$$

**Proposition D.3.** *Given $d$-sparse vectors $u \in \mathbb{R}_{\geq 0}^m, v \in \mathbb{R}_{\geq 0}^n$ and query access to $\mathbf{A} \in \mathbb{R}^{m \times n}$ with $\|u\|_1 = \|v\|_1 = 1$ and $0 \leq A_{i,j} \leq 1$, we can estimate $u^\mathsf{T} \mathbf{A} v$ within additive error $\varepsilon$ with success probability at least $1 - \delta$ in randomized time $O(d + \log(1/\delta)/\varepsilon^2)$.*

*Proof.* We regard $u$ and $v$ as probability distributions. Consider the following procedure.

1. Sample a row $i \in [m]$ from the probability distribution $u$, and a column $j \in [n]$ from the probability distribution $v$;

2. Output $X = A_{i,j}$.

Then,

$$\mathbb{E}[X] = \sum_{i \in [m]} \sum_{j \in [n]} u_i v_j A_{i,j} = u^\mathsf{T} \mathbf{A} v.$$

We repeat the above procedure for $k = \lceil \ln(2/\delta)/2\varepsilon^2 \rceil = \Theta(\log(1/\delta)/\varepsilon^2)$ times, and let $\bar{X}$ be the mean value of the outputs. Then, by Theorem D.2, we have

$$\Pr[|\bar{X} - u^\mathsf{T} \mathbf{A} v| \leq \varepsilon] \geq 1 - 2\exp(-2k\varepsilon^2) \geq 1 - \delta.$$

We note that after a simple $O(d)$-time deterministic pre-processing procedure according to $u$, we can generate a sample from the probability distribution $u$ in randomized $O(1)$ time. Therefore, we can estimate $u^\mathsf{T} \mathbf{A} v$ within additive error $\varepsilon$ with success probability at least $1 - \delta$ in randomized time $O(d + \log(1/\delta)/\varepsilon^2)$. □