# OpenReview forum: "Quantum Speedups in Linear Programming via Sublinear Multi-Gibbs Sampling"
_ICLR.cc/2024/Conference — Submitted to ICLR 2024_

### Official Review · Reviewer_8VsV · 2023-10-13

**Soundness:** 3 good
**Presentation:** 2 fair
**Contribution:** 3 good
**Rating:** 5
**Confidence:** 3

**Summary:**

In this paper, the authors presented an improved multi-level Gibbs sampler using quantum machinery that improves the zero-sum game solver of [BGJST23] from $O(\sqrt{m+n}\gamma^{2.5})$ to $O(\sqrt{m+n}\gamma^{2.25})$. Consequentially, this also leads to an improved linear program solver that runs in $O(\sqrt{m+n}\gamma^{2.25})$ time. Here, $\gamma=Rr/\epsilon$ where $\epsilon$ is the accuracy and $R, r$ are $\ell_1$ radius of optimal solution respectively. They allow $\epsilon=O((m+n)^{-2})$ instead of the $\epsilon=O((m+n)^{-1})$ due to [BGJST23]. To obtain these results, the authors leverage a better polynomial approximation due to [SV14] and a reparametrization of preprocessing. This enables them to improve [BGJST23].

**Strengths:**

Improving the dependence on $\gamma$ is important for quantum zero-sum game. This also leads to improvement on linear program solver. It is worth noting that second-order method such as interior point method is less meaningful as the dependence on $\epsilon^{-1}$ is still polynomial due to reading out solutions from quantum state.

**Weaknesses:**

The techniques in this paper are not novel, they are relatively standard adaptation of approaches such as better polynomial approximation. On the polynomial approximation front, I suspect a recent result due to Aggarwal and Alman [AA22] would lead direct improvement over the polynomial degree. The multi-Gibbs sampling algorithm appears to be novel, unfortunately, there're very little explanations on why the algorithm works and where the improvements actually come from. The algorithms and main results are listed in the paper without further elaboration, leaving the paper hard to evaluate and grasp the actual contribution. Nevertheless, I still believe the results obtained are strong, but in my honest opinion, the presentation can be significantly improved.

[AA22] A. Aggarwal and J. Alman. Optimal-Degree Polynomial Approximations for Exponentials and Gaussian Kernel Density Estimation. CCC'22.

**Questions:**

Some questions:

1. In your Table 1, both [BGJST23] and [GJLW23] has the same runtime. Are there some differences in terms of other parameters? What are the differences between these two results?

2. Why does Algorithm 1, 2 and 3 work? What's the intuition behind that? What are the major differences between these algorithms and prior works? I think it's important to provide a high-level overview of the approaches, the key innovations and the steps in the algorithms that are worth highlighting.

A few typos/citation corrections:

1. In the 4th paragraph of introduction (page 1), the improvement of LP solver to $O(n^{2+1/18})$ is due to [JSWZ21] not [CLS19].

2. In the paragraph above Theorem 1.2 (page 2), [BGJST23] requires $1/\epsilon=O(m+n)$ but not $O((m+n)^{-1})$.

---

### Official Review · Reviewer_m7tP · 2023-10-20

**Soundness:** 3 good
**Presentation:** 1 poor
**Contribution:** 2 fair
**Rating:** 5
**Confidence:** 4

**Summary:**

This paper studies the quantum algorithm for solving matrix zero-sum games and linear programming, which are fundamental problems in optimization theory with numerous real-world applications. The main result is an $\tilde{O}(\sqrt{m+n}/\epsilon^{2.25})$-time quantum solver for matrix zero-sum games. It improves the previous state-of-the-art quantum algorithm by a factor of $(1/\epsilon)^{1/4}$. As a corollary, it implies an $\tilde{O}(\sqrt{m+n}/(Rr\epsilon)^{2.25})$-time quantum linear programming solver. Technically, the algorithms are based on the sample-based optimistic weight update framework. And these improvements mainly come from a new multi-Gibbs sampler, which can (approximately) generate $k$ samples from the Gibbs distribution of the form $e^{\beta H}$, where $H$ is a diagonal Hamiltonian (i.e., diagonal matrix). Notably, all previous quantum Gibbs sampler has a linear dependence on $\beta$. This paper overcomes this barrier via an average argument.

**Strengths:**

This is a solid theory paper, improving previous algorithms’ time complexities (Bouland et al., 2023; Gao et al., 2023). Furthermore, the new technical tool (multi-Gibbs sampler) improved in this paper is also quite interesting and can potentially be used to solve other sampling and optimization problems.

**Weaknesses:**

First and foremost, the writing of this paper is really bad. It is highly non-self-contained and directly uses theorem/lemma statements from previous papers without any explanation. In the main text it basically just puts the pseudocodes of the algorithms and states the time complexities, which makes it extremely difficult to understand how these algorithms work and to justify the correctness of the results. It only proves the Gibbs sampling result and does not prove the main theorems for matrix zero-sum games and LP. In short, the writing is not publication-ready yet and requires major revision. Second, I think the improvements over previous works are not significant enough to make it a new paper. It heavily uses technical results (Gao et al., 2023). For the LP result, it is a little hard to justify its usefulness since it is a first-order method and still has a $(Rr/\epsilon)^{2.25}$ factor. This paper would be strengthened if it could provide examples of practical problems such that using the new quantum LP solver can improve the classical methods.

**Questions:**

Page 1: “In Cohen et al. (2021), the authors improve the latter complexity to …” The reference here should be (Jiang et al., 2021).

Page 1, line -10: pure strategy and mixed strategy are used but not defined.

Page 2: “matrix zero-sum games” are discussed on this page, while “matrix games” are used on the previous page. Are they referring to the same object? If so, they should be unified to avoid confusion.

Page 2: “which is by designing a quantum Gibbs sampler and use the framework proposed in Grigoriadis & Khachiyan (1995).” use -> using

Page 2: “in van Apeldoorn & Gilyen (2019b), the authors gave a quantum algorithm ´ specifically for matrix zero-sum games and linear programming …” It would be better to state the complexities of their matrix game algorithm and LP algorithm separately.

Page 2: “All the previous quantum Gibbs samplers used in van Apeldoorn & Gilyen (2019b); Bouland et al. (2023); Gao et al. (2023) have a linear dependence on the ´ ℓ1-norm β of the vector u” This part is too technical and not self-contained. It will make the non-expert readers very difficult to follow.

Thm. 1.1: before stating the result, it would be better first informally to define the zero-sum matrix game and its Nash equilibrium.

Thm. 1.3: this theorem uses a lot of terms that are not defined yet. E.g., what is the Gibbs distribution, and what are the inputs to the algorithm? It’s also important to mention that the output samples are classical.

Thm. 1.3: why does it require a lower bound and an upper bound for $k$? Why the algorithm cannot output fewer samples?

Sec. 1.2, line 6: what is $u_{\max}$?

Page 4, first paragraph: what’s the meaning of $\xi$?

Page 4, input model: the oracle $\mathcal{O}_{A}’$ requires that each entry of $A$ is in $[0,1]$?

Thm. 2.2: to output $|\phi_0\rangle$ exactly, does the algorithm need to know the value of $p$?

Thm. 2.3: it would be better to explain the difference between the consistent version and the usual version of amplitude estimation before stating the theorem. $f$ is not defined. Is it some fixed function? So, when $p$ and $s$ are fixed, with high probability, the algorithm's output will be the same?

Sec. 2.4, line 1: “Quantum singular value transformation is a powerful quantum algorithm design framework proposed in Gao et al. (2023).” The reference is incorrect. QSVT has been developed for more than five years!

Page 6, line -2: “The concept of amplitude-encoding is proposed in Gao et al. (2023).” This is not accurate. The idea of encoding classical data in the amplitudes of a quantum state has been developed for decades. Def. 2.3 is not essentially different from previous works.

Alg. 1: what does the XOR mean? The first input of the procedure FindMax is an oracle. However, AmpEst is an algorithm that outputs a number. And what does line 4 AmpEst()|i>|0> mean?

Page 7, Thm. 3.1: it should mention where is the proof of the theorem.

Page 7 - 8: As the main part of this paper, more discussions about the idea of the new Gibbs sampling algorithm should be provided instead of just putting the pseudocodes and the theorems in the main text. It is extremely difficult to understand and justify the results!

Sec. 4.2 & 4.3: no intuition for the algorithm and no explanation for the theorem’s proof! These subsections fail to deliver new information not in the introduction.

Appendix A gives a detailed introduction to the consistent amplitude estimation. However, it is never mentioned in the main text.

Thm. A.1: this theorem should cite some previous work on QPE. And in the last line, “time. gates” -> “time and gates”.

Did Thm. A.3 and A.4 proved in Gao et al. (2023)? If so, it should be pointed out in the paper.

Lem. C.2: how to guarantee that each $w_i$ is in $[-1,1]$?

Lem. C.3, proof: $u_{\min}$ is not defined.

Trace distance/norm is used several times in this paper but never defined.

I cannot find the proof of Cor. 4.2 and Cor. 4.3 in the appendix.

In Cor. 4.2, the algorithm has $T$ iterations, and in each iteration, it uses $T$ Gibbs samples. Thus, shouldn’t the time complexity be $T\cdot T_{Gibbs}$? In Cor. 4.2, it is just $T_{Gibbs}$. I think it is a typo, and the result should be correct.

Thm. D.1: it shows that the value of the matrix zero-sum game can induce the value of LP. Does the quantum LP solver only output the value? Is there a reduction from the matrix game solution to the LP solution?

---

### Official Review · Reviewer_Q6ko · 2023-11-01

**Soundness:** 4 excellent
**Presentation:** 4 excellent
**Contribution:** 3 good
**Rating:** 6
**Confidence:** 4

**Summary:**

The paper provides a quantum algorithm that can output an epsilon-approxmation of the Nash equilibrium for a zero-sum game. While the runtime matches the previous known results, the range of epsilon for which this algorithm can work is "quadratically" bigger than the previous results. In other words, using the current algorithm a better approximation can be output.

Their algorithm can also be extended to obtain an epsilon-approximation for a LP with same time complexity in worst case but able to handle smaller epsilons. Their algorithm is based on the multiplication weight technique.

**Strengths:**

The technical contribution of this paper is quite good.
The paper is also very well written.

**Weaknesses:**

While theoretically the paper is really impressive, the practical applicability is not very clear. Particularly, because it talking about quantum algorithms. In this respect a more theoretical conference would be possible more suitable for the paper.

**Questions:**

Nil

---

### Meta-Review · Area_Chair_k8nm · 2023-12-07

**Metareview:**

This submission aims to improve quantum algorithms for solving linear programming and zero-sum games, achieving a quantum speedup by using a novel quantum multi-Gibbs sampler. The paper received mixed reviews. All reviewers acknowledge the technical strength of the paper; the improvement in dependence on certain parameters will be important for quantum zero-sum game solvers. However, the novelty is unclear. Reviewer m7tP criticizes the poor presentation and lack of self-contained explanations. They note the paper's heavy reliance on previous works and lack of significant improvements over them. Reviewer 8VsV notes that the techniques used are not novel, and the presentation lacks sufficient explanation of why the algorithms work and their improvements over previous works. Reviewer Q6ko suggests that a more theoretical conference might be a better fit for this work.

**Justification For Why Not Higher Score:**

Reviewers found the paper technically strong but criticized its lack of clarity and accessibility, as it made it challenging to grasp the full significance and potential impact of the work.

**Justification For Why Not Lower Score:**

N/A

---

### Decision · Program_Chairs · 2024-01-16

Reject